# A Novel Method of Small Object Detection in UAV Remote Sensing Images Based on Feature Alignment of Candidate Regions

**Jinkang Wang** , **Faming Shao ***, **Xiaohui He** and **Guanlin Lu**

College of Field Engineering, Army Engineering University of People's Liberation Army of China, Nanjing 210007, China
* Correspondence: shaofaming@aeu.edu.cn; Tel.: +86-185-4985-4591

**Abstract:** To solve the problem of low detection accuracy of small objects in UAV optical remote sensing images due to low contrast, dense distribution, and weak features, this paper proposes a small object detection method based on feature alignment of candidate regions is proposed for remote sensing images. Firstly, AFA-FPN (Attention-based Feature Alignment FPN) defines the corresponding relationship between feature mappings, solves the misregistration of features between adjacent levels, and improves the recognition ability of small objects by aligning and fusing shallow spatial features and deep semantic features. Secondly, the PHDA (Polarization Hybrid Domain Attention) module captures local areas containing small object features through parallel channel domain attention and spatial domain attention. It assigns a larger weight to these areas to alleviate the interference of background noise. Then, the rotation branch uses RROI to rotate the horizontal frame obtained by RPN, which avoids missing detection of small objects with dense distribution and arbitrary direction. Next, the rotation branch uses RROI to rotate the horizontal box obtained by RPN. It solves the problem of missing detection of small objects with dense distribution and arbitrary direction and prevents feature mismatch between the object and candidate regions. Finally, the loss function is improved to better reflect the difference between the predicted value and the ground truth. Experiments are conducted on a self-made dataset. The experimental results show that the mAP of the proposed method reaches 82.04% and the detection speed reaches 24.3 FPS, which is significantly higher than that of the state-of-the-art methods. Meanwhile, the ablation experiment verifies the rationality of each module.

**Keywords:** UAV remote sensing images; small object; feature alignment; polarization hybrid domain attention

## 1. Introduction

UAV optical remote sensing images are collected by sensors in the visible light band (0.38~0.76 microns). They can directly reflect detailed information such as the shape, color, and texture of ground objects, thus facilitating direct observation of human beings. Optical remote sensing image object detection refers to using a specific method to search and mark interesting objects from images, such as airplanes, tanks, ships, and vehicles. Optical remote sensing object detection is a basic technology that plays an important role in analyzing remote sensing images, and it is an important basis for urban planning, land use, and traffic diversion [1].

UAV remote sensing images cover a large area of the ground, and the object size is generally small. There are two difficulties in the detection of small objects in remote sensing images: (1) Lack of feature information. Because small objects occupy fewer pixels in the image and are surrounded by complex background information, it is difficult for the network to extract effective features of small objects, which affects the subsequent positioning and recognition tasks. (2) Difficulty of positioning. Compared with large

objects, the positive sample matching rate of small objects is low, so small objects do not contribute much to the network training, which affects the ability of the network to locate the small objects.

To solve the problem that it is difficult to detect small objects in UAV remote sensing images, this paper focuses on the effects of hierarchical features, attention mechanism, and rotating bounding box on the detection performance of small objects. To improve the feature expression of small objects, the model aligns and fuses the shallow spatial features and deep semantic features. Based on this, double attention is adopted to denoise the remote sensing image. Then, the rotating bounding box is introduced to alleviate the missing detection problem of small objects with a dense distribution and arbitrary direction and a mismatch between the object and the candidate region features. Finally, the loss function is improved to enhance the robustness of the model. The contributions of this paper are summarized as follows:

1.  APA-FPN establishes the corresponding relationship between feature mappings, thus solving the misalignment between features and enriching the semantic representation of shallow features of small objects.
2.  PHDA captures local areas containing small object features through parallel channel domain attention and spatial domain attention. These areas are assigned with a larger weight to eliminate the interference of background noise.
3.  The rotation branch introduces the rotating bounding box to solve the missing detection of small objects with dense distribution and arbitrary direction and the mismatch between the object and the candidate region features.
4.  The loss function is improved to enhance the robustness of the model. The experimental results show that the proposed method is more accurate and faster than the state-of-the-art remote sensing object detection methods.

## 2. Related Works

UAV remote sensing image detection based on deep learning is increasingly mature, and various methods have been proposed to improve the detection performance [2–4]. Remote sensing small objects are difficult to extract because they occupy only a few pixels, and the information is easily lost under layer-by-layer convolution calculation. How to improve the detection performance of small objects in remote sensing images has become a research hotspot.

### 2.1. Conventional Object Detection

According to whether ROI (region of interest) extraction is needed, the object detection methods based on deep learning can be divided into two-stage detection and one-stage detection.

The two-stage object detection method has high accuracy, but it involves additional computation, and the speed cannot meet the real-time requirement. Typical two-stage object detection methods are represented by R-CNN [5,6] series. RFCN [7] improves the network after the ROI pooling layer and replaces the full connection with the location-sensitive score map obtained by full convolution, which greatly improves the detection speed. Mask-RCNN [8] combines segmentation and detection tasks, and it uses interpolation to align ROI, which further improves detection accuracy. FPN [9] uses a top-down architecture with a horizontal connection, which can extract strong semantic information at all levels. One-stage object detection completes the whole detection process in one stage, which is fast and can meet the requirements of real-time systems, but the detection accuracy is slightly lower than that of two-stage object detection. Typical one-stage object detection methods are the YOLO series [10–13] and SSD [14]. Additionally, RetinaNet [15] combines FPN and FCN [16] networks and proposes an improved cross-entropy focal loss, which effectively eliminates the category imbalance problem. CornerNet [17] adopts a corner detection method to avoid generating anchor boxes, and it introduces the concept of corner pooling to better locate corners, which effectively improves the detection performance.

## 2.2. Remote Sensing Object Detection

The above-mentioned general object detection methods have achieved great progress in recent years, and they are effective for conventional object detection tasks. However, there are some problems in remote sensing images, such as complex background, arbitrary-oriented objects, densely distributed small objects, etc., which pose a great challenge to object detection in multi-scene remote sensing images.

R$^3$Det [18] integrates a feature refinement module that can realize the fine distinction of object features, and it adopts a more accurate rotation loss. Based on this, it can achieve a mAP of 76.47 on the DOTA data set. Yang et al. [19] employ CSL to solve the problem of boundary discontinuity, and it transforms the regression problem of angle prediction into a classification problem, which effectively alleviates the influence of remote sensing object rotation in any direction and improves detection performance. GWD [20] uses two Wasserstein distances with an arbitrary Gaussian distribution as the rotation IoU loss, which effectively solves the boundary discontinuity and square-like problems. ReDet [21] adds the rotation equivariant network to the detector to extract rotation equivariant features, which can accurately predict the direction and significantly reduce the model size; meanwhile, it integrates RiRoI Align to adaptively extract rotation invariant features according to the direction of RoI. Chen et al. proposed PIoU loss [22], which calculates IoU by pixel-by-pixel judgment and improves the rotation angle and IoU accuracy at the same time. S$^2$A-Net [23] first refines the initial anchor into a rotating anchor and then adjusts the position of feature sampling points to extract the aligned depth features under the guidance of the refined anchor box, which achieves good detection results. Oriented R-CNN [24] designs a directed candidate frame generation network, called Oriented RPN, which achieves good detection results and has an execution speed comparable to that of single-stage object detection.

## 2.3. Conventional Small Object Detection

Aiming at the bottleneck of low performance of small object detection, more experts and scholars are turning their attention to the field of small object detection. At present, a series of effective improvement methods have been proposed [25–27].

PANet [28] adds a bottom-up path enhancement branch to FPN [9], which more fully integrates the information of high-level features and low-level features. MDSSD [29] adds a low-level feature map and a high-level feature map element by element to SSD, and the generated feature map is used to predict objects of different scales. Zhang et al. [30] fused feature maps with different expansion rates to learn multi-scale information of the object, which improved the detection accuracy without increasing the computational complexity. Couplenet [31] combines local information, global information, and context information to improve the detection accuracy of small objects. Li et al. [32] combined top-down and bottom-up attention mechanisms and applied a circular flow to guide feature extraction, which makes the network better positioned on the object features and achieves high-precision small object detection. Xi et al. [33] introduced a pairwise constraint to describe the semantic similarity and used the context information of candidate objects to improve the detection performance of small objects. The above method improves the detection accuracy of small objects by feature fusion, multi-scale prediction, attention mechanism, and context-based information, and the detection results are improved.

## 2.4. Remote Sensing Small Object Detection

Remote sensing images have many characteristics, such as large texture differences, low contrast, arbitrary dense distribution, and ultra-fine, which make it more difficult to detect small objects. Many effective network frameworks have been proposed for multi-scale object detection.

PIoU [34] and SCRDet++ [35] have been proposed to verify the effectiveness of denoising methods in optical remote sensing small object detection. The former uses an attention mechanism, while the latter uses instance-level denoising on a feature map to highlight and enhance the features of small objects, thus detecting small and messy remote sensing

objects. Li et al. [36] proposed an enhanced architecture for small object detection, which reduces the times of network downsampling to avoid information loss in the subsequent transmission process. Meanwhile, they fused shallow features and deep features to enrich the small object information in the feature map. Hou et al. [37] proposed a fusion strategy of cascading features, which integrates shallow position information and deep semantic information to fuse all layers of features and enhance the cascading effect. Qu et al. [38] designed an efficient feature fusion network by adopting atrous convolution, which enhances the effective receptive field of deep features. Zhang et al. [39] aggregated the context information in the shallow feature map and increased the training weight of the small objects, which improves the detection performance. Since the scale of small objects in real remote sensing images may be different, it is difficult for the feature fusion module to splice feature maps into deep features. Therefore, only using a feature fusion module may become the bottleneck of detection. The existing studies have verified the effectiveness of combining an attention mechanism with a feature fusion module in detecting remote-sensing small objects [40]. This paper selects some representative works and briefly introduces their methods, advantages, and disadvantages in Table 1.

**Table 1.** Comparison of semantic image segmentation methods.

| Algorithm | Brief Methodology | Highlights | Limitations |
|---|---|---|---|
| CSL | Boundary problem transformation, Circular Smooth Label | The regression problem of angle is transformed into a classification problem, and the range of prediction results is limited to eliminate large boundary loss. | Too many angle categories will cause the head part of RetinaNet to be too thick, resulting in low calculation efficiency. |
| $R^3$Det | Pixel by pixel feature interpolation, Rotation RetinaNet | The feature refinement module realizes feature reconstruction and alignment through pixel-by-pixel feature interpolation. An approximate SkewIoU loss is proposed to achieve more accurate rotation estimation. | The continuity of local pixel information is interrupted, and the adaptability to unknown deformation is poor. |
| MDSSD | Feature fusion, multi-scale convolution | The high-level feature map with rich semantic features is enlarged by deconvlution, and then fused with the low-level feature map to achieve more refined feature extraction. | Without designing the rotating frame, it is impossible to accurately predict the rotating object. |
| $S^2A-Net$ | Anchor refinement network, Active rotating filter | FAM generates high-quality anchors through an anchor thinning network, and adaptively aligns convolution features. ODM uses an active rotation filter to encode the direction information, and generates the characteristics of direction sensitivity and direction invariance. | Insensitive to small-scale remote sensing objects, and the detection effect is not good. |
| PIoU | PIoU loss | PIoU loss is calculated by pixel-by-pixel judgment and is continuously differentiable, which effectively improves the detection effect of inclined targets. | Insensitive to small-scale remote sensing objects. |
| SCRDet++ | Instance-Level Feature Denoising, Rotation Loss Smoothing | In feature map, a novel instance-level denoising module is designed to suppress noise and highlight prospects, and an improved Smooth L1 loss is designed to solve the boundary problem of rotating bounding box regression. | Without feature alignment, the ability of feature expression is poor. |
| Oriented R-CNN | Oriented RPN, Oriented RCNN head | Orientation RPN generates high-quality orientation proposals at almost no cost. Directional R-CNN header refines and identifies the region of interest. | The detection effect of small and chaotic objects is not good. |

Based on the advantages and disadvantages of the above studies, this paper proposes a method for UAV remote sensing small object detection. AFA-FPN aligns and fuses the shallow spatial features and deep semantic features. Based on this, PHDA introduces dual attention to denoise the remote sensing image. Then, the rotating bounding box is adopted to alleviate the missing detection problem of small objects with dense distribution and arbitrary direction and the mismatch between the object and the candidate region features. Finally, the loss function is improved to enhance the robustness of the model. To verify the effectiveness of the proposed method for UAV remote sensing small object detection, this paper extracts a certain proportion of small object images from three published datasets and reorganizes them into a new dataset called RSSO. The experimental results on the RSSO dataset show that compared with the state-of-the-art methods, our method has an obvious effect on improving the detection accuracy and speed of remote sensing small objects.

## 3. Overview of the Proposed Method

The structure of the remote sensing small object detection model proposed in this paper is shown in Figure 1. ResNet-101 [41] is used as the backbone network to extract features, and then AFA-FPN clarifies the relationship between feature mappings to solve the problem of misregistration of features between adjacent levels and ensure that the deep semantic features of small objects can be well transmitted to the shallow feature map. PHDA captures the local areas containing small object features in the tensor of the feature map by parallel channel domain attention and spatial domain attention, and it assigns a larger weight to these areas to eliminate the interference of background noise. The rotation branch adopts a two-stage rotation regression to reduce the regression difficulty of the rotation candidate box, extract rotation-invariant regional features, and eliminate the mismatch between regional features and objects. Finally, the loss function is improved to obtain better classification and regression results.

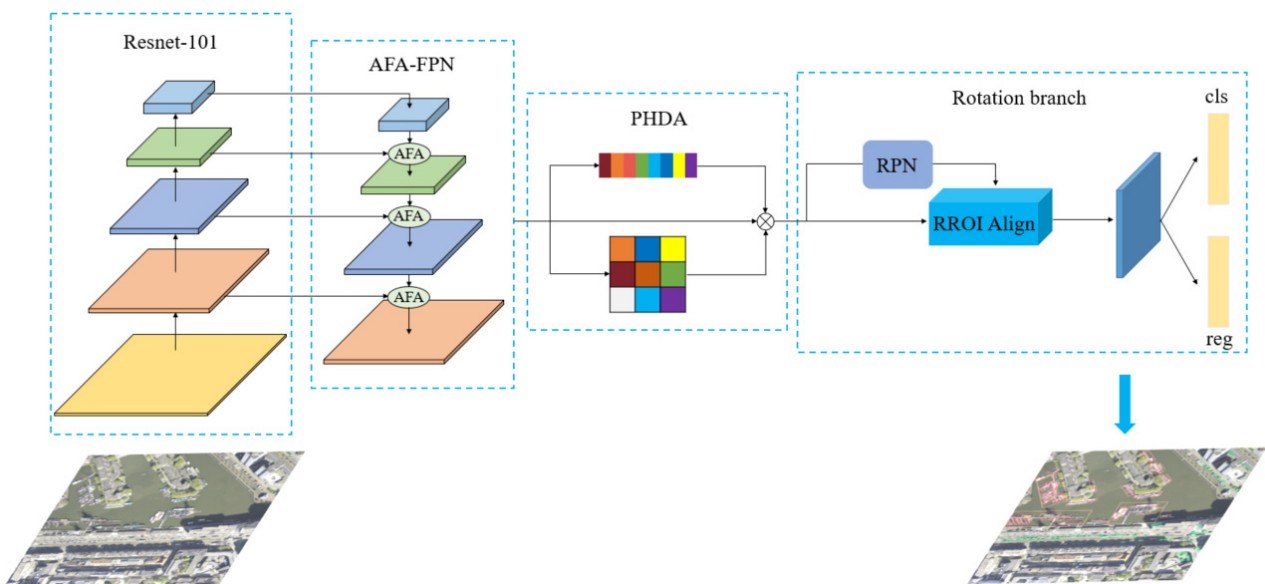

**Figure 1.** The pipelines of the proposed method.

### 3.1. AFA-FPN (Attention-Based Feature Alignment FPN Module)

FPN was proposed to improve the performance of multi-scale object detection tasks, and it can also improve the detection performance of small objects. Many methods use FPN to fuse features of small objects, and good results are obtained [42–44]. However, there are still some problems that affect the detection performance of small objects in actual detection tasks, and the most prominent problem is the feature fusion between levels. The traditional FPN network simply reduces the channel number of the deep feature map

through $1 \times 1$ convolution layers and then performs upsampling to make the deep feature map the same size as the adjacent shallow feature map. Finally, deep semantic features and shallow spatial features are fused through element-wise addition. This method improves the multi-scale detection performance of the detector, but its heuristic mapping mechanism between the pyramid level and the candidate box size makes small objects share the same feature map with normal or large objects, which seriously limits the ability of the shallow feature map to learn the features of the small objects. In addition, upsampling the deep feature map can easily cause the semantic features in the adjacent level feature maps to be misaligned so that the small object features are submerged in the complex background.

Based on the above analysis, this subsection introduces a feature alignment module called AFA-FPN to explicitly establish the correspondence between feature maps to solve the misalignment between features. By predicting the flow field, the misregistration between features of adjacent layers can be avoided, which ensures that the deep semantic features of small targets can be well transmitted to the shallow feature map, thus effectively solving the problem of small target information loss during the convolution process. The structure of AFA-FPN is shown in Figure 2.

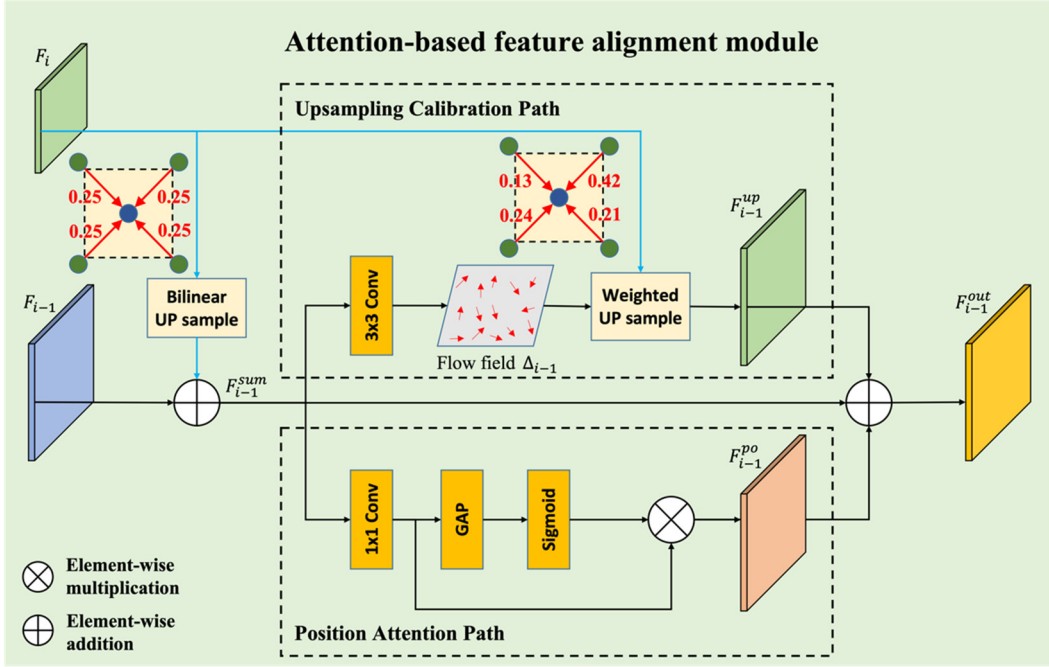

**Figure 2.** The structure of the AFA-FPN module.

Firstly, the deep and shallow feature maps are adjusted by a convolutional layer to the same channel size. Then, the deep feature maps are upsampled and added with the shallow feature map. The resulting feature maps are:

$$F_{i-1}^{sum} = Conv_i\big(f_{up}(F_i)\big) + Conv_{i-1}(F_{i-1}) \tag{1}$$

where $F_i$ and $F_{i-1}$ $i \in (3, 4, 5)$ are deep and shallow feature maps, respectively, and $F_{i-1}$ has a larger spatial size than $F_i$; $Conv_i$ and $Conv_{i-1}$ are convolution operations for channel adjustment of deep and shallow feature maps, respectively, and $f_{up}$ is the upsampling operation for deep feature maps. The fused features $F_{i-1}^{sum}$ are fed into the upsampling calibration path and object attention branch, respectively.

**Upsampling Calibration Path.** In this path, a $3 \times 3$ convolution kernel is used to predict the feature offsets:

$$\Delta_{i-1} = Conv_\Delta\big(F_{i-1}^{sum}\big) \tag{2}$$

where $Conv_\Delta$ is the convolution operation for predicting the feature offset of the deep feature maps; $\Delta_{i-1} \in R^{H_{i-1} \times W_{i-1} \times 4}$ is the predicted offset field, and it is used to weighted upsample the $F_i$.

The $\Delta_{i-1}$ has four values at each spatial location $\Delta_{i-1} = (w_{i-1}^{ul}, w_{i-1}^{ur}, w_{i-1}^{lr}, w_{i-1}^{ll})$, which represents the weights of four adjacent pixel points: the upper left point, the upper right point, the lower right point, and the lower left point. The upsampled feature map is represented as:

$$F_{i-1}^{up} = w_{i-1}^{ul} F_i^{ul} + w_{i-1}^{ur} F_i^{ur} + w_{i-1}^{lr} F_i^{lr} + w_{i-1}^{ll} F_i^{ll} \tag{3}$$

where $F_i^{ul}, F_i^{ur}, F_i^{lr}, F_i^{ll}$ represent the $F_i$ features of different upsampled locations. This operation is similar to the traditional bilinear upsampling operation, except that the traditional upsampling has $w_{i-1}^{ul} = w_{i-1}^{ur} = w_{i-1}^{lr} = w_{i-1}^{ll} = 0.25$, while $\Delta_{i-1}$ is adaptively calculated for the weighted upsampling.

object **Attention Path.** A $1 \times 1$ convolutional layer is first used to process the input feature maps $F_{i-1}^{sum}$, and then the position weight is calculated through the global average pooling (GAP) operation. Finally, the *Sigmoid* activation is applied to normalize the extracted features. The output can be expressed as:

$$F_{i-1}^{po} = \sigma\big(\theta\big(Conv_{1 \times 1}(F_{i-1}^{sum})\big)\big) \times F_{i-1}^{sum} \tag{4}$$

where $Conv_{1 \times 1}$ is the $1 \times 1$ convolutional layer, $\theta(\cdot)$ is realized through GAP, and $\sigma(\cdot)$ is the *Sigmoid* function.

**Feature Alignment.** Finally, the feature maps $F_{i-1}^{sum}$, $F_{i-1}^{up}$ and $F_{i-1}^{po}$ are fused for feature alignment, and they are from the identity mapping, upsampling calibration path, and object attention path, respectively. The resulting feature is

$$F_{i-1}^{out} = F_{i-1}^{sum} + F_{i-1}^{up} + F_{i-1}^{po} \tag{5}$$

The flow chart of the AFA-FPN module is shown in Figure 3.

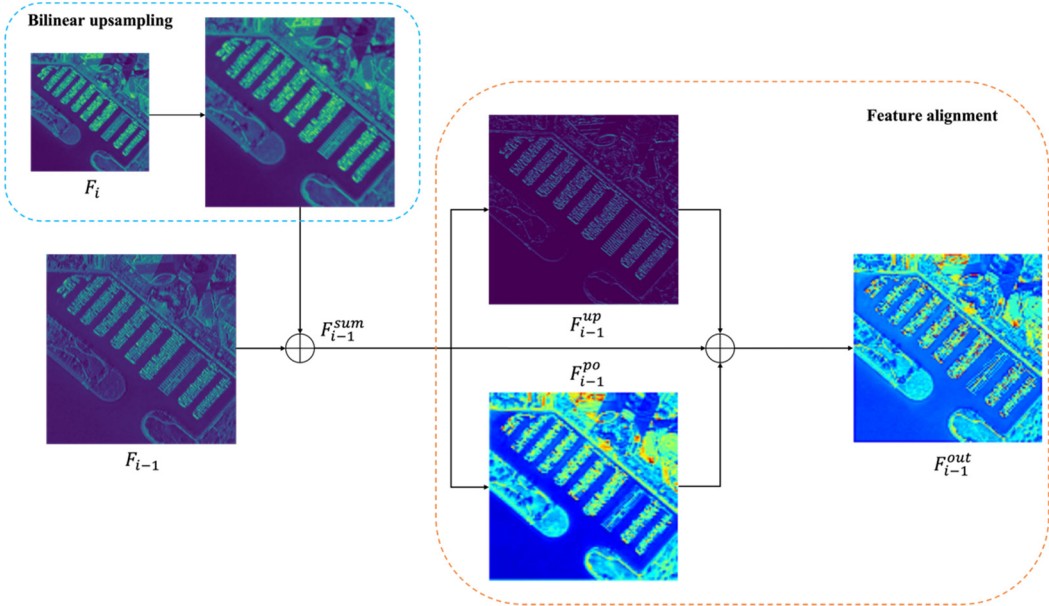

**Figure 3.** The flow chart of the AFA-FPN module.

Generally speaking, $F_{i-1}^{up}$ is the feature map generated by the upsampling calibration path, where the object boundary is upsampled in a high quality. $F_{i-1}^{po}$ is the feature map generated by the position attention path, where the objects are highlighted. As can be seen from Figure 3, in the upsampled feature map obtained by the AFA-FPN module,

the deep semantic features converge to certain positions of the corresponding objects in the shallow feature map. This helps to solve the problem of small object feature loss in the subsequent fusion process. Meanwhile, it enriches the semantic representation of the shallow features of small objects so that the features of small objects will not be submerged in the deep background noise. Thus, AFA-FPN provides a good feature basis for the subsequent localization and classification of small objects, thus effectively improving the detection performance of small objects.

### 3.2. PHDA (Polarization Hybrid Domain Attention Module)

Remote sensing images contain many complex and dense small objects, and some large-scale scene objects such as ground field tracks, baseball fields, etc. The huge difference in object scale and the few small object pixels lead to serious miss detection of small objects. Adopting an attention mechanism to pay special attention to remote sensing small objects and suppressing background noise has become a hot spot in small object detection [45–47].

The attention mechanism is essentially a set of weighting factors, which can emphasize regions of interest while suppressing unrelated background regions in a "dynamic weighting" approach. Compared with CNN, RNN, and other network structures, the attention mechanism module has less complexity and fewer parameters, so it requires less computing power. Meanwhile, the attention module can be calculated in parallel, and the calculation of each step of the attention mechanism does not depend on the calculation result of the previous step, so it can be processed in parallel like CNN. The general attention mechanism does not involve a polarization function and only gives cursory attention to the object that needs attention. For the detection of small objects in UAV images, the general attention mechanism cannot achieve good results due to their small size, various types, and different directions. Therefore, this paper introduces the polarization hybrid domain attention (PHDA) mechanism to improve the detection effect of small objects, as shown in Figure 4.

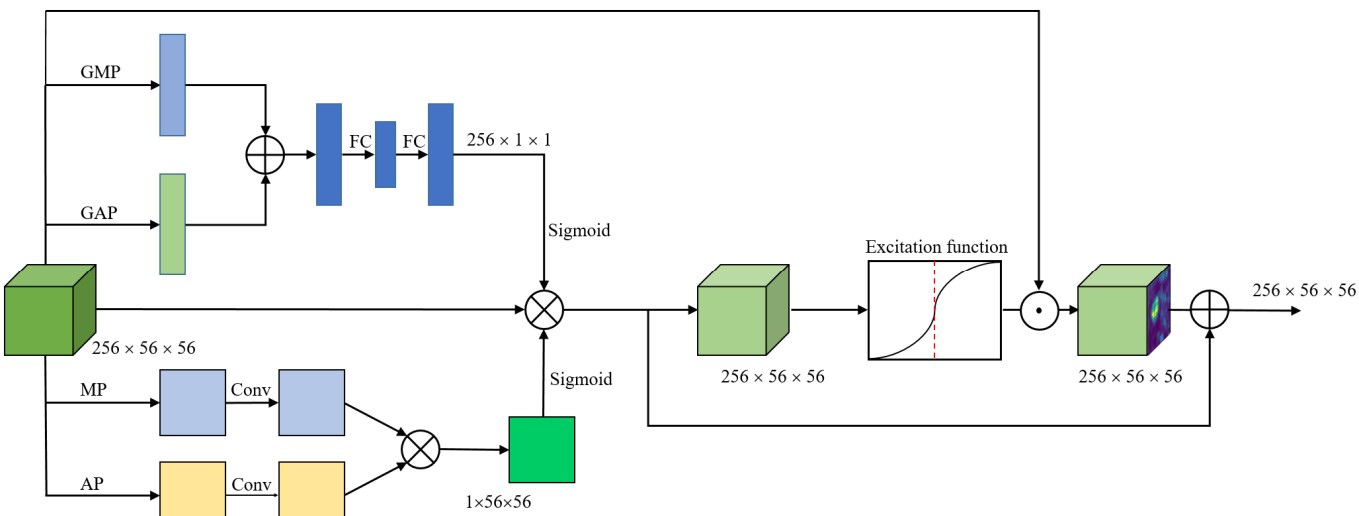

**Figure 4.** The structure of the PHDA module.

Channel attention aims to find the correlation between different channels (feature maps), automatically obtain the importance of each feature channel through network learning, and give each channel a different weight coefficient to strengthen important features and suppress unimportant features. However, the channel-based attention module ignores the location information, so it is necessary to introduce the spatial attention module to enhance the spatial location features. The adoption of an attention mechanism can extract location information, reduce the interference of background information, and generate a weight mask for each location, thus improving the network detection performance.

For a given input feature $T \in R^{w \times h \times c}$, the channel attention map $H_C$ extracts the weight of each channel through global average pooling, global maximum pooling, and full connection. The process can be expressed as follows:

$$H_C(T) = \sigma(W_2(W_1(T_{gap} + T_{gmp}))) \tag{6}$$

where $T_{gap}$ and $T_{gmp}$, respectively, represent the feature graphs obtained by global average pooling and global maximum pooling, $W_2 \in R^{C/r \times C}$ and $W_1 \in rR^{C \times C/r}$ represent the weights of all the connected layers, and $\sigma$ represents the *Sigmoid* function.

Correspondingly, the spatial attention $H_S$ extracts the spatial position weight of the input feature pixels through average pooling and maximum pooling. Maximum pooling means to maximize the pixels in the pool area, and the feature map obtained in this way is more sensitive to texture feature information. Average pooling means to average the images in the pool area, and the feature information obtained in this way is more sensitive to the background information. Spatial attention mainly focuses on extracting the spatial position weight of the input feature pixels, which does not require high background accuracy. Therefore, asymmetric $3 \times 1$ and $1 \times 3$ convolutions are used to compress and accelerate the model in average pooling to reduce the number of parameters. For maximum pooling, the traditional $3 \times 3$ convolution is employed to extract texture feature information, and its expression is as follows:

$$H_S(T) = \sigma(c^{3 \times 3}(T_{ap}) + cat(c^{1 \times 3}, c^{3 \times 1})(T_{mp})) \tag{7}$$

where $c^{3 \times 3}$, $c^{1 \times 3}$, and $c^{3 \times 1}$ represent the convolution with a convolution kernel of $3 \times 3$, $1 \times 3$, and $3 \times 1$, respectively, *cat* represents serial operation, and $T_{ap}$ and $T_{mp}$ represent the feature graph obtained by performing average pooling and maximum pooling on the input features, respectively.

The output of the PHDA module is as follows:

$$\begin{aligned} H &= H_C(T) \otimes H_S(T) \otimes T \\ T' &= H + \psi(\sigma(H)) \odot T \end{aligned} \tag{8}$$

where $\otimes$ and $\odot$ represent the tensor product and element-wise multiplication, respectively, $\sigma$ represents the *Sigmoid* function, and $\psi$ represents the polarization function. To increase the attention more towards the small object area and reduce the interference of unimportant background information [48], this paper uses the following excitation function:

$$\psi(x) = \frac{1}{1 + e^{-\eta(x - 0.5)}} \tag{9}$$

where $\eta$ is the adjustment factor for controlling the activation intensity, and it is set to 10. It can be seen from Equation (9) that irrelevant areas with an attention weight of less than 0.5 are greatly suppressed to reduce noise interference to small object areas, while important areas with an attention weight of greater than 0.5 are highlighted to enhance small object features. The introduction of the polarization function further enhances the effect of the attention mechanism and highlights the characteristic areas of small objects.

### 3.3. Rotation Branch

The rotation branch is divided into two stages: the first stage is RRoI alignment, and the second stage is RRoI rotation position-sensitive pooling. RRoI alignment is to obtain the offset of rotation ground truths (RGTs) relative to horizontal RoI; RRoI rotation position-sensitive pooling is to further refine the rotation area to keep the rotation invariant properties [49].

### 3.3.1. RRoI Alignment

RRoI alignment is mainly responsible for learning RRoIs from the horizontal ROI feature maps. Assume that $n$ horizontal ROIs are obtained as $\{\xi_i\}$. This paper uses $(x, y, w, h)$ to represent the position, width, and height of horizontal RoI. In an ideal situation, each horizontal RoI is the outer rectangle of the RRoI. Therefore, this paper infers the geometric shape of RRoI from each feature map with the full connection layer.

$$
\begin{aligned}
t_x^* &= \tfrac{1}{w_r}\left((x^* - x_r)\cos\theta_r + (y^* - y_r)\sin\theta_r\right) \\
t_y^* &= \tfrac{1}{h_r}\left((y^* - y_r)\cos\theta_r - (x^* - x_r)\sin\theta_r\right) \\
t_w^* &= \log\tfrac{w^*}{w_r}, t_h^* = \log\tfrac{h^*}{h_r} \\
t_\theta^* &= \tfrac{1}{2\pi}\left((\theta^* - \theta_r)\bmod 2\pi\right)
\end{aligned}
\tag{10}
$$

In Equation (10), $(x_r, y_r, w_r, h_r, \theta_r)$ represent the position, width, height, and direction of the RRoI, respectively. $(x^*, y^*, w^*, h^*, \theta^*)$ are the information parameters of the ground truth of the OBB (oriented bounding box). The mod operation is used to adjust the angle offset within $[0, 2\pi)$. As shown in Equation (11), the fully connected layer outputs a vector $(t_x, t_y, t_w, t_h, t_\theta)$ for each feature map.

$$
t = \varsigma(\zeta; \kappa)
\tag{11}
$$

where $\varsigma$ represents the fully connected layer, and $\kappa$ is the weight parameter of $\varsigma$.

### 3.3.2. RRoI Rotation Position-Sensitive Pooling

After obtaining the parameters of the RRoI, the RRoI rotation position-sensitive pooling operation extracts rotation-invariant features of the oriented objects.

Given a feature map $D$ of shape $(H, W, K \times K \times C)$ and RRoI $(x_r, y_r, w_r, h_r, \theta_r)$, where $(x_r, y_r)$ is the center of the RRoI, $(w_r, h_r)$ is the width and height of the RRoI, and $\theta_r$ represents the direction of the RRoI. This paper divides RRoI into $K \times K$ blocks. For block $d_{i,j,c}$ with channel $c(0 \le c < C)$ and index $(i, j)(0 \le i, j < K)$, its output $Y$ can be expressed as follows:

$$
Y_c(i, j) = \frac{\sum\limits_{(x,y)\in B_{i,j}} d_{i,j,c}(\Gamma_\theta(x, y))}{n_{ij}}
\tag{12}
$$

where $B_{i,j}$ represents the coordinate set, $n_{ij}$ is the number of all pixels in the interval, and $(x, y) \in B_{i,j}$ is the global coordinate of the feature point $P_{i,j}$. $(x, y) \in B_{i,j}$ can be converted into $(x', y')$ by $\Gamma_\theta$, and the transformation equation is shown as follows, which is realized by bilinear interpolation:

$$
\begin{pmatrix} x' \\ y' \end{pmatrix} = \begin{pmatrix} \cos\theta & -\sin\theta \\ \sin\theta & \cos\theta \end{pmatrix} \begin{pmatrix} x - \tfrac{w_r}{2} \\ y - \tfrac{h_r}{2} \end{pmatrix} + \begin{pmatrix} x_r \\ y_r \end{pmatrix}
\tag{13}
$$

Floating-point parameters are reserved in the quantization process of rotation-sensitive pooling, then the pixel value of the object is obtained by bilinear interpolation, and finally the output is obtained by maximum pooling. Taking $k = 3$ as an example, nine RROI feature maps are spliced into an RROI voting map. Then, the RRoIs are voted through nine position-sensitive scores to obtain the feature map that keeps rotating position sensitivity on different position blocks. The specific process is shown in Figure 5.

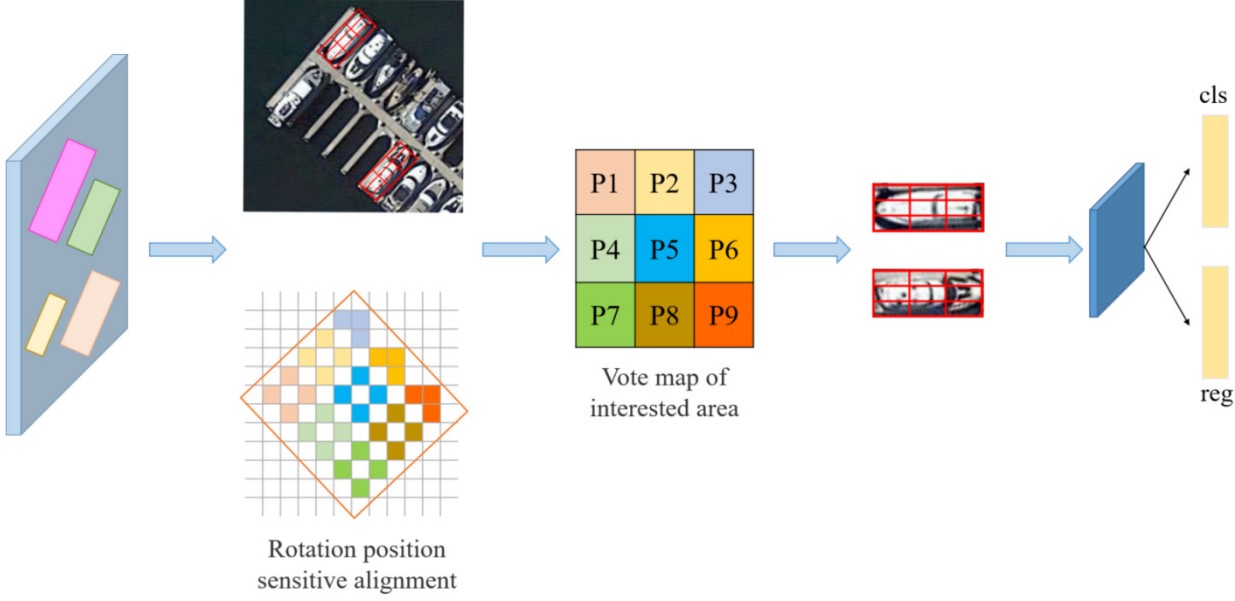

**Figure 5.** The flow chart of RRoI rotation position-sensitive pooling.

Figure 6 shows the two-stage box rotation process of the rotation branch. In the first stage, the blue box is initially aligned to the green box, and in the second stage, the green box is further refined to the red box. GT-H and PR-H represent the ground truth and predicted value of the horizontal box, respectively, and FS-R and SS-R represent the rotating box obtained in the first stage and the second stage, respectively.

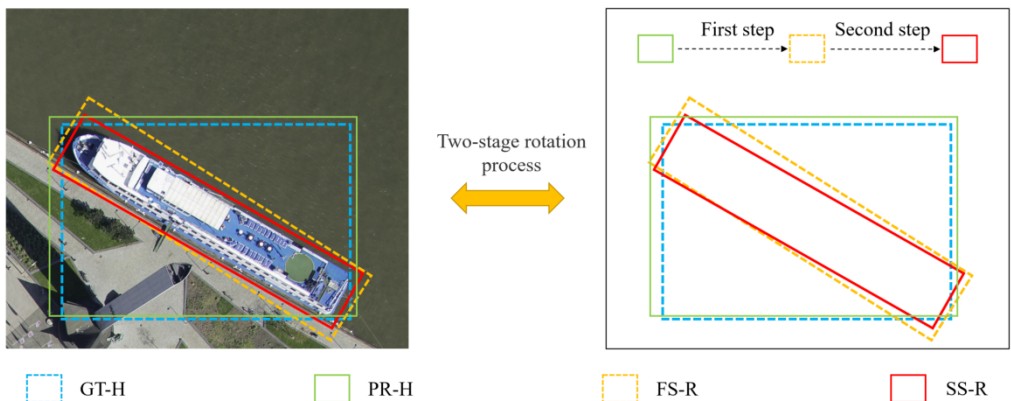

**Figure 6.** The schematic diagram of the two-stage box rotation process of the rotation branch.

*3.4. Loss Function*

3.4.1. Classification Loss Function

This paper adopts the focal loss function [15] as the classification loss function, as shown in Formula (14).

$$FL = \begin{cases} -(1 - \hat{P})^{\rho} \log(\hat{P}), & if \ y = 1 \\ -\hat{P}^{\rho} \log(1 - \hat{P}), & if \ y = 0 \end{cases} \tag{14}$$

where $\hat{P} \in (0, 1)$ represents the predicted value, and $\rho > 0$ is the adjustable factor.

Let $P_t$ represent the probability that the predicted category is a real label, which is defined in Formula (15). After substituting $P_t$ into Formula (14), the focal loss function can be written in Formula (16).

$$P_t = \begin{cases} \hat{P}, \; if \; y = 1 \\ 1 - \hat{P}, \; else \end{cases} \tag{15}$$

$$L_{cls} = FL = -(1 - P_t)^\rho \log(P_t) \tag{16}$$

It can be seen from Formula (16) that the focal loss function is the optimization of the cross-entropy loss function. Compared with the cross-entropy loss function, the focal loss function has one more modulation factor $(1 - P_t)^\rho$, and when $\rho = 0$, the focal loss becomes a cross-entropy loss.

The main idea of the loss function is to add weight to positive and negative samples, measure the loss contribution of hard samples and easy samples, and adjust the weight of the loss contribution of the positive and negative samples. However, many candidate frames will be generated during the detection process, and the number of objects in an image is limited. This means that most of the candidate frames are negative samples, especially for small objects. The optimized focal loss function can solve this problem well.

3.4.2. Regression Loss Function

Compared with the methods that use GIoU [50] as the regression loss function, this paper adopts the more comprehensive CIoU [51] as the regression loss function. The optimization object of GIoU is the overlapping area of the predicted box and the real frame, which helps to solve the problem that IoU is 0. Although it can reflect the directional difference of anchor boxes to a certain extent, when the intersection value of two boxes is the same, it cannot reflect the specific intersection situation. In addition to the overlapping area, the CIoU loss function also considers distance, scale, and aspect ratio. It is defined as follows:

$$v = \frac{4}{\pi^2} \left( \arctan \frac{w^{gt}}{h^{gt}} - \arctan \frac{w}{h} \right)^2 \tag{17}$$

$$CIoU = IoU - \frac{\rho^2 (b, b^{gt})}{c^2} - \alpha v \tag{18}$$

$$L_{reg} = L_{CIoU} = 1 - CIoU \tag{19}$$

where $\alpha$ is the weight, $v$ is used to measure the similarity of aspect ratio, $b$ and $b^{gt}$ are prediction box and ground truth, respectively, $w$ and $h$ are the width and height of the prediction box, respectively, $w^{gt}$ and $h^{gt}$ are the width and height of the ground truth, respectively, $c$ is the diagonal of the minimum circumscribed moment of two rectangular boxes, $\rho$ is the Euclidean distance between the center points of two rectangular boxes, $CIoU$ is a more comprehensive optimization of $IoU$, and $L_{CIoU}$ is the loss of $CIoU$.

The classical RPN loss proposed by Faster RCNN [6] is used as the RPN loss in this paper, as shown in the following formula, where the meaning of each letter is the same as that of Faster RCNN.

$$L_{RPN} = \frac{1}{N_{cls}} \sum_i L_{cls}(p_i, p_i^*) + \frac{1}{N_{reg}} \sum_i p_i^* L_{reg}(t_i, t_i^*) \tag{20}$$

The total loss function can be expressed as:

$$L = \frac{1}{N} \sum_{n=1}^{N} L_{cls} + \frac{\lambda_1}{N} \sum_{n=1}^{N} L_{reg} + L_{RPN} \tag{21}$$

where $L_{cls}$ and $L_{reg}$ represent classification loss and regression loss, respectively; $N$ represents the number of candidate boxes; and $\lambda$ is a weight balance parameter, and it is set to 1.

## 4. Experimental Analysis

### 4.1. Dataset

To verify the effectiveness of the proposed method, this paper conducts experiments on a self-made remote sensing small object database RSSO. The images of RSSO are taken from NWPU VHR-10 [52], DOTA [53], and UCAS AOD [54]. NWPU VHR-10 is a public dataset released by Northwestern Polytechnical University for remote sensing image object detection. This dataset includes 800 remote sensing images of 10 types of ground objects, such as harbors, bridges, and vehicles. The DOTA dataset contains 2809 remote sensing images of 15 types of ground objects, such as planes, ships, and storage tanks, and it is the largest and most comprehensive remote sensing image data set at present. UCAS AOD is used for aircraft and vehicle detection. Almost all objects in this dataset are small objects. The 600 aircraft images contain 3210 aircraft, and the 310 vehicle images contain 2819 vehicles.

This paper defines the object whose length and width are less than 0.1 of the original graphic size as a small object, the object whose length and width are greater than 0.1 but less than 0.3 of the original graphic size as a medium object, and the object whose length and width are greater than 0.3 of the original graphic size as a large object. According to this definition, this paper selects the images that mainly contain small objects from the above three datasets, and finally 400, 1000, and 600 images are obtained from NWPU VHR-10, DOTA, and UCAS AOD, respectively. After cropping and scaling, the dataset is expanded to 4000 images and called the RSSO dataset. The objects in our dataset are of different sizes, but they are mostly small-sized objects, with various types and arbitrary orientations. Figure 7 shows some images in the dataset. The RSSO dataset is divided into a training set, a validation set, and a test set at the ratio of 0.6, 0.15, and 0.25.

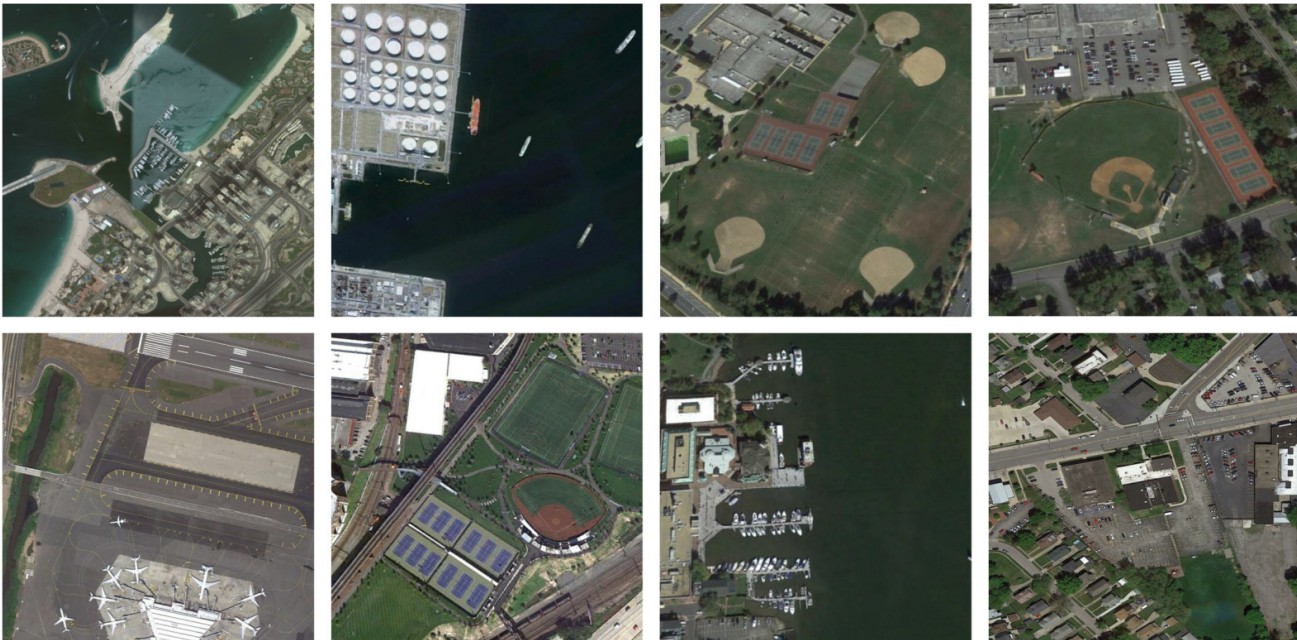

**Figure 7.** Some image examples from the RSSO dataset.

### 4.2. Experimental Setting

This paper uses Ubuntu 18.04 as the software experimental platform to verify the designed network structure and uses NVIDIA GeForce RTX 3090Ti as the graphics processing unit in both training and testing. Experiments are conducted on PyTorch. In the network training, the SGD optimizer with momentum is used to train the model; the momentum parameter is set to 0.9, the initial learning rate is set to 0.01, and it decays to 1/10 every

50,000 iterations. Additionally, the weight decay coefficient is 0.0005, and the batch size is 32. The total number of training iterations for the RSSO dataset is 200,000.

In this paper, the evaluation indicators are Recall (recall rate) and Precision (precision rate), and they are defined in Formulas (22) and (23). The discriminant condition of TP (True Positive) and FP (False Positive) is whether the IoU between the object identified as a positive sample and the ground truth region is greater than the specified threshold. FN (False Negative) refers to omitting the ground truth that has not been detected.

$$R = \frac{TP}{TP + FN} = \frac{TP}{all\ ground\ truth} \tag{22}$$

$$P = \frac{TP}{TP + FP} = \frac{TP}{all\ \text{detect}} \tag{23}$$

Generally, the precision rate and the recall rate are mutually restricted, and $AP$ can achieve a balance between the two. Therefore, $AP$ is often used as the standard for evaluating the network detection ability. The P-R curve can be drawn by taking the $R$ value as the abscissa and the $P$ value as the ordinate, and then $AP$ is the integral area enclosed by the P-R curve and the coordinate axis. The expression of $AP$ is shown in Formula (24). $mAP$ is the average of the $AP$ values of each category, which represents the average detection accuracy of the method for the whole dataset. The expression of $mAP$ is shown in Formula (25).

$$AP = \int_0^1 P(R)\mathrm{d}R \tag{24}$$

$$mAP = \frac{1}{Q}\sum_{q=1}^{Q} AP(q) \tag{25}$$

where $P(R)$ represents the precision when the recall rate on the PR curve is R, $q$ represents a certain object category, and $Q$ represents the total number of object categories.

### 4.3. Ablation Experiment

To verify the effectiveness of the proposed module, the ablation experiment analyzes its effectiveness in improving the method's performance. Table 2 shows the influence of FPN, AFA-FPN, PHDA module, rotation branch, and loss function on the model detection effect. In this table, $AP_S$ indicates the average detection accuracy of small objects, and $mAP@0.5$ refers to the average AP value when the IoU threshold is 0.5. $mAP@0.5 : 0.95$ represents the average accuracy when the IoU threshold is increased from 0.5 to 0.95 at a step size of 0.05.

**Table 2.** Results of ablation experiment. The best results are in bold.

| Baseline | FPN | AFA-FPN | PHDA | Rotation Branch | GIoU | CIoU | $AP_S$ | $mAP@0.5$ | $mAP@0.5:0.95$ |
|---|---|---|---|---|---|---|---|---|---|
| ✓ | ✓ | | | | ✓ | | 48.75 | 64.28 | 53.27 |
| ✓ | | ✓ | | | ✓ | | 53.60 | 67.59 | 54.03 |
| ✓ | | ✓ | | | | ✓ | 56.21 | 66.37 | 54.82 |
| ✓ | ✓ | | | | | ✓ | 52.74 | 65.98 | 53.94 |
| ✓ | | ✓ | ✓ | | ✓ | | 68.92 | 73.57 | 56.05 |
| ✓ | | ✓ | | ✓ | | ✓ | 64.31 | 72.11 | 55.71 |
| ✓ | | ✓ | ✓ | ✓ | ✓ | | 74.48 | 77.14 | 58.45 |
| ✓ | | ✓ | ✓ | ✓ | | ✓ | **78.21** | **82.04** | **59.70** |

It can be seen from the table that compared with FPN, our proposed AFA-FPN improves $AP_S$ by 4.85%, which greatly improves the detection accuracy of small objects. Meanwhile, $mAP@0.5$ and $mAP@0.5 : 0.95$ are increased by 3.31% and 0.76%, respectively, which indicates that AFA-FPN is suitable for improving the detection accuracy of medium and large objects. Additionally, the use of PHDA and the rotation branch brings great benefits to the performance of small object detection, and the average detection accuracy

is greatly improved. Compared with GIOU, CIOU is more comprehensive, with faster convergence and better results. The model combining AFA-FPN, PHDA, rotation branch, and CIOU achieves the best detection effect, with $AP_S$ reaching 78.21%, and $mAP@0.5$ and $mAP@0.5 : 0.95$ reaching 82.04% and 59.70%, respectively. These results prove the rationality and superiority of the model designed in this paper.

　　To highlight the good effect on improving the detection performance of small objects, this paper compares the P-R curves for detecting small, medium, large objects, and all objects. The comparison result is shown in Figure 8, where sequence numbers 1 to 7 represent the first to seventh rows of Table 1, respectively. From Figure 8, it can be seen that our proposed method can improve the detection accuracy of small objects the most. Additionally, the detection accuracy of medium, large and all objects is slightly improved. The P-R curve verifies the effectiveness of our proposed method for remote sensing small object detection, which is consistent with the results obtained in Table 1.

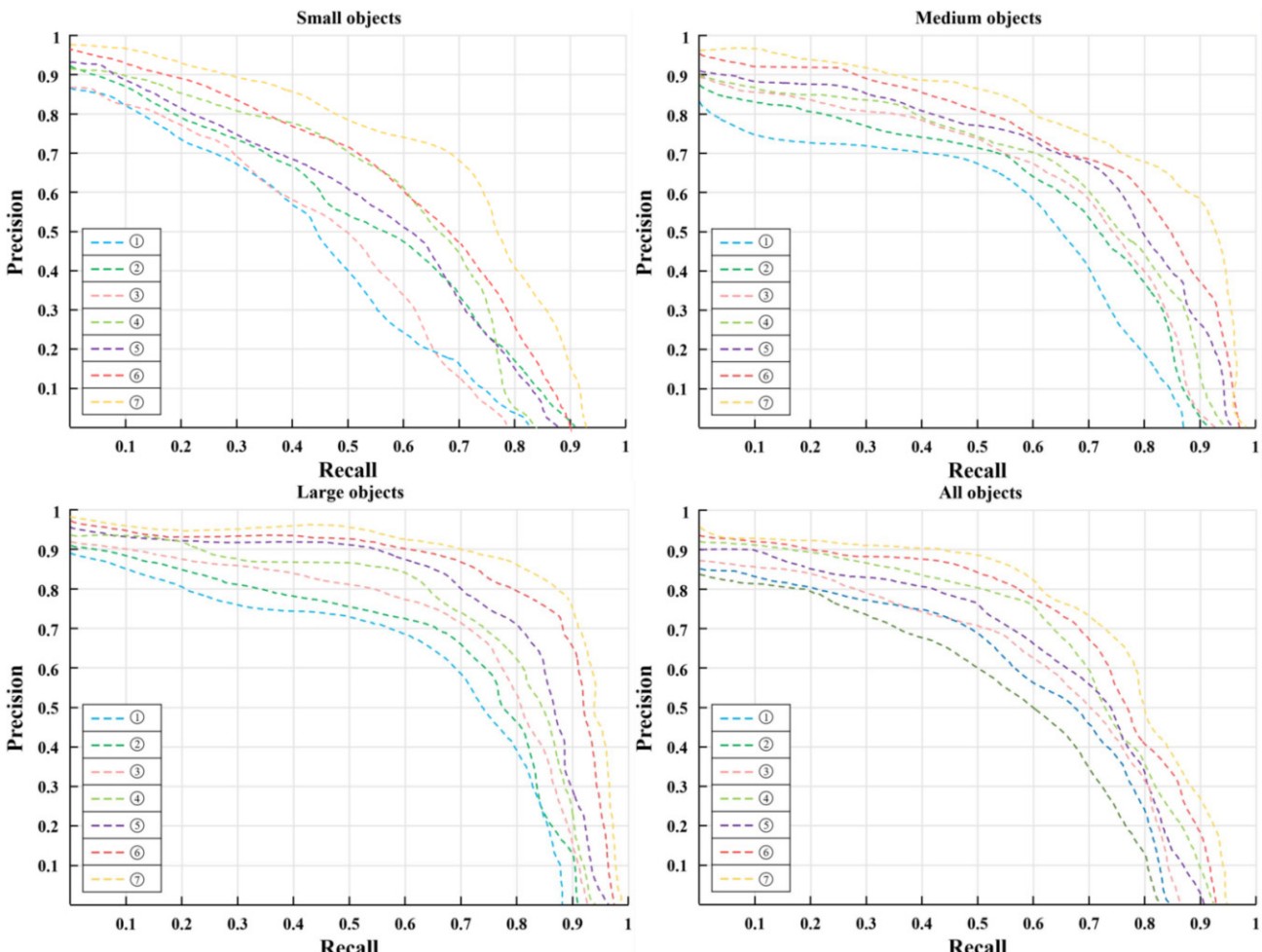

**Figure 8.** Comparison of P-R curves of small, medium, and large objects and all objects.

　　Based on this, this paper chooses several backbone networks with all the modules proposed in this paper to conduct experiments on the RSSO dataset to verify the rationality of choosing Resnet-101 as the backbone network. The selected networks are VGG-16 [55], VGG-19 [55], ResNet-50 [38], ResNet-101 [38], and Alex-Net [56]. The corresponding precision–recall curve is shown in Figure 9. The results of Figure 9 indicate that Resnet-101 achieves better precision and recall rates than other backbone networks.

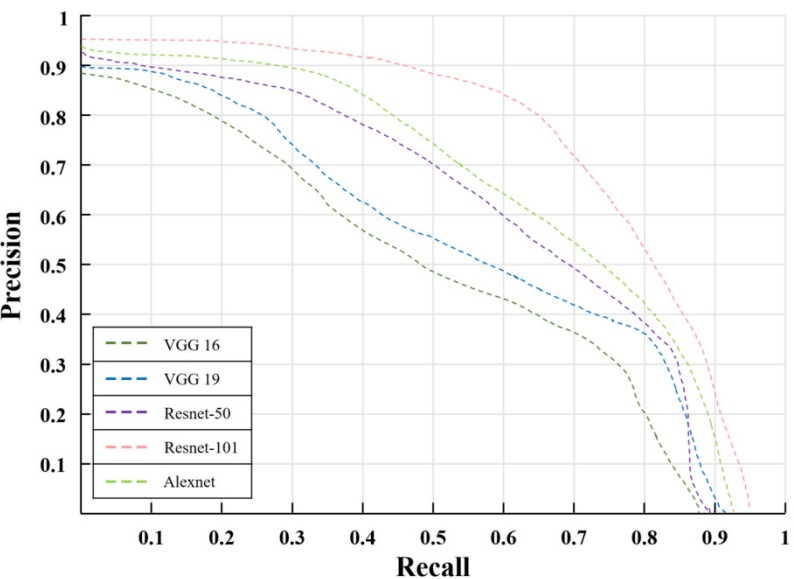

**Figure 9.** Comparison of the P-R curves of adopting different backbone networks.

### 4.4. Experimental Results and Analysis

To verify the effectiveness of our model structure, Table 3 presents the detection results of CSL, R³Det, MDSSD, S²A − Net, PIoU, SCRDet++, Oriented R-CNN, and our proposed methods for various objects. For the convenience of expression, the object categories in the table are abbreviated as PL-Plane, BD-Baseball diamond, ST—Storage tank, SH—Ship, TC- Tennis court, BR—Bridge, GTF—Ground field track, HA—Harbor, BC—Basketball court, SV—Small vehicle, LV—Large vehicle, SP—Swimming pool, SBF—Soccer ball field, HC—Helicopter, and RA—Roundabout.

**Table 3.** Experimental results of our proposed method compared with the state-of-the-art methods on the RSSO dataset. The best results are in bold.

| Method | PL | BD | ST | SH | TC | BR | GTF | HA |
|---|---|---|---|---|---|---|---|---|
| CSL | 67.38 | 69.80 | 66.15 | 71.65 | 83.61 | 65.75 | 77.15 | 76.53 |
| R³Det | 68.59 | 69.37 | 71.36 | 68.24 | 86.24 | 64.38 | 82.43 | 70.52 |
| MDSSD | 62.11 | 71.25 | 71.54 | 70.35 | 85.34 | 66.54 | 79.01 | 72.77 |
| S²A − Net | 75.96 | 76.89 | 75.38 | 76.77 | 88.48 | 74.68 | 84.28 | 79.85 |
| PIoU | 78.43 | 79.54 | 78.09 | 78.05 | 89.71 | 78.02 | 84.21 | 80.64 |
| SCRDet++ | 76.78 | 77.42 | 76.08 | 76.91 | **90.46** | 77.25 | **87.92** | **83.60** |
| Oriented R-CNN | 80.93 | **80.42** | 81.23 | 78.94 | 89.60 | **78.17** | 85.14 | 80.53 |
| Ours | **82.42** | 79.46 | **82.21** | **79.02** | 89.94 | 77.63 | 85.47 | 81.75 |

| Method | BC | SV | LV | SBF | RA | SP | HC | mAP |
|---|---|---|---|---|---|---|---|---|
| CSL | 68.52 | 61.06 | 58.08 | 79.22 | 74.75 | 78.98 | 76.21 | 71.66 |
| R³Det | 67.44 | 63.21 | 61.19 | **84.68** | 76.88 | 76.54 | 72.64 | 72.25 |
| MDSSD | 75.07 | 69.52 | 71.44 | 78.55 | **89.42** | 83.59 | 79.05 | 75.04 |
| S²A − Net | 77.26 | 66.24 | 71.07 | 78.22 | 81.79 | 86.87 | 77.96 | 79.04 |
| PIoU | 78.11 | 67.83 | 70.22 | 80.06 | 82.56 | 87.98 | 82.58 | 78.57 |
| SCRDet++ | 70.84 | 64.32 | 71.03 | 82.60 | 86.71 | 88.19 | 84.01 | 79.61 |
| Oriented R-CNN | 78.58 | 68.55 | 70.08 | 81.35 | 84.56 | **88.02** | 84.37 | 80.24 |
| Ours | **79.27** | **74.11** | **75.16** | 83.49 | 88.31 | 87.58 | **84.81** | **82.04** |

From Table 3, it can be seen that the proposed method achieves the highest detection accuracy for small-scale planes, ships, oil tanks, small vehicles, and helicopters. Meanwhile, the detection accuracy of medium-scale objects such as large vehicles and basketball courts is also the highest. Although the detection accuracy of large-scale objects such as ground

field tracks and soccer ball fields is not the highest, it is also among the highest of the comparative advanced methods. The MAP of the proposed method reaches 82.04%, which is increased by 1.8%–10.38% compared with that of the state-of-the-art detection methods. This verifies the effectiveness of the proposed method for detecting remote sensing small objects. Compared with other methods, AFA-FPN makes the relationship between feature maps clear and enhances the feature expression of small objects. PHDA eliminates the interference of background noise to some extent, which is significant for detecting small objects. The rotation branch and improved loss function enable the network model to obtain better classification and regression results. Generally, the proposed method enhances and improves the detection of small objects in remote sensing images and obtains obvious results.

In addition, to more comprehensively evaluate the superiority of our proposed method, Table 4 presents the accuracy comparison of different size objects, the model accuracy comparison of different IOU thresholds, and the speed comparison with CSL, $R^3$Det, MDSSD, $S^2A-Net$, PIoU, SCRDet++, Oriented R-CNN, Yolo v4 and Yolo v7. $AP_S$, $AP_M$, and $AP_L$ represent the average detection accuracy of small, medium, and large objects in the dataset, respectively. $mAP@0.5$, $mAP@0.75$, and $mAP@0.9$ refer to the average AP value when the IoU threshold is 0.5, 0.75, and 0.9, respectively. FPS stands for Frames Per Second.

**Table 4.** Comparison of the object detection performance on the RSSO dataset. The best results are in bold.

| Method | $AP_S$ | $AP_M$ | $AP_L$ | $mAP@0.5$ | $mAP@0.75$ | $mAP@0.9$ | FPS |
|---|---|---|---|---|---|---|---|
| CSL | 61.02 | 71.02 | 80.23 | 70.22 | 58.11 | 46.97 | 22.0 |
| $R^3$Det | 63.21 | 72.38 | 81.54 | 71.35 | 58.64 | 47.82 | 17.3 |
| MDSSD | 68.85 | 75.61 | 77.69 | 71.98 | 59.07 | 48.33 | **33.4** |
| $S^2A-Net$ | 71.58 | 76.71 | 78.09 | 75.84 | 60.49 | 49.01 | 22.3 |
| PIoU | 74.02 | 78.04 | 80.14 | 76.35 | 61.50 | 51.20 | 20.7 |
| SCRDet++ | 70.99 | 79.09 | **81.30** | 75.08 | 62.44 | 49.03 | 21.2 |
| Oriented R-CNN | 75.54 | 81.99 | 80.24 | 78.45 | 64.94 | 51.96 | 18.6 |
| Yolo v4 | 39.54 | 48.09 | 58.22 | 56.47 | 41.98 | 33.84 | 106 |
| Yolo v7 | 48.31 | 57.87 | 69.61 | 61.29 | 55.31 | 39.05 | 114 |
| Ours | **78.21** | **84.77** | 80.43 | **82.04** | **69.70** | **53.41** | 24.3 |

It can be seen from Table 4 that the proposed method achieves the highest detection accuracy for small and medium objects, and the improvement of small objects is greater. Compared with the state-of-the-art methods, the accuracy is improved by 2.67%–17.19%. Meanwhile, the average detection accuracy for each IOU threshold range is the highest, which means that the proposed method can delineate more accurate bounding boxes for remote sensing image object detection. For densely distributed remote sensing small objects, a more accurate bounding box can alleviate the impact of dense distribution and improve detection accuracy. Although the calculation speed of the proposed method is not the fastest, it basically meets the requirements of real-time performance. Compared with the traditional Yolo v4 and Yolo v7, the improvement is even greater. Although the Yolo series algorithms have a faster speed, because they are not designed for the object detection of remote sensing small objects, they are insensitive to remote sensing small objects and their detection accuracy is not high. Thus, our method performs the best for remote sensing small object detection both in detection accuracy and speed.

Figures 10–13 show the detection effects of different methods on the RSSO dataset under the conditions of simple backgrounds, complex backgrounds, multi-scale objects, and dense small objects, respectively. Our method is compared with CSL, $R^3$Det, MDSSD, $S^2A-Net$, PIoU, SCRDet++, and Oriented R-CNN. It can be seen that MDSSD does not have a rotating frame design, so the prediction boxes are all horizontal. The prediction boxes of the other methods are rotating boxes. From a subjective point of view, our method has a stronger detection ability for small objects and can effectively reduce the omission ratio of small objects under the conditions of a simple background, complex background, multi-scale objects, or dense small objects. Due to the two-stage regression of the anchor

box by the rotation branch, the prediction box of the proposed method is the closest to the ground truth, and the IOU between the prediction box of the proposed method and the ground truth is the highest among all methods.

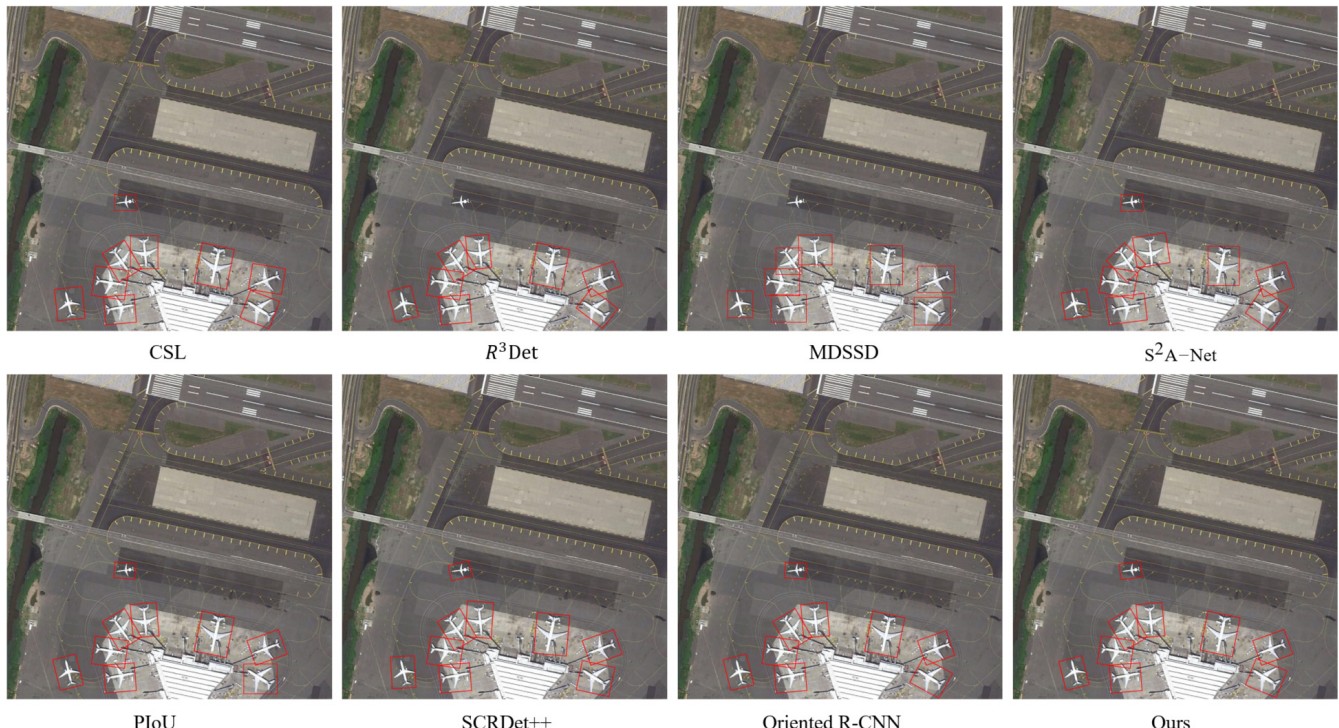

**Figure 10.** Visualization of the detection performance between our proposed method and the state-of-the-art methods in simple background.

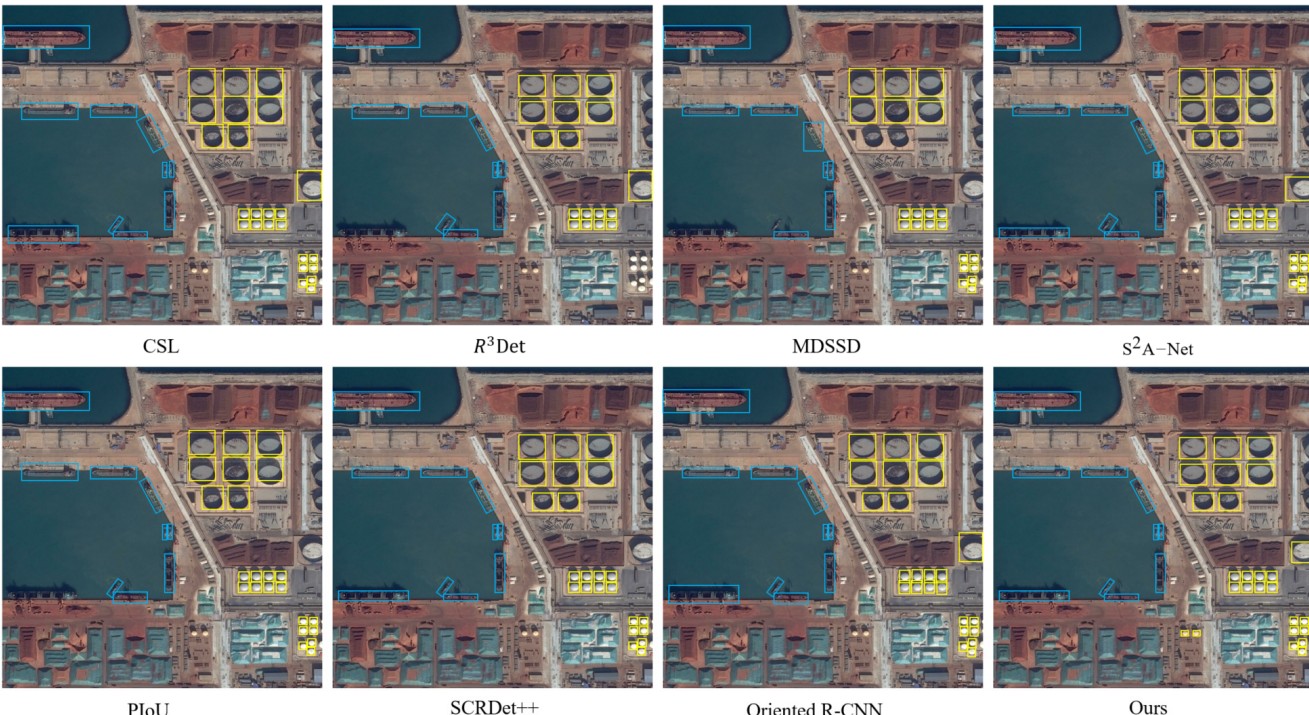

**Figure 11.** Visualization of the detection performance between our proposed method and the state-of-the-art methods in a complex background.

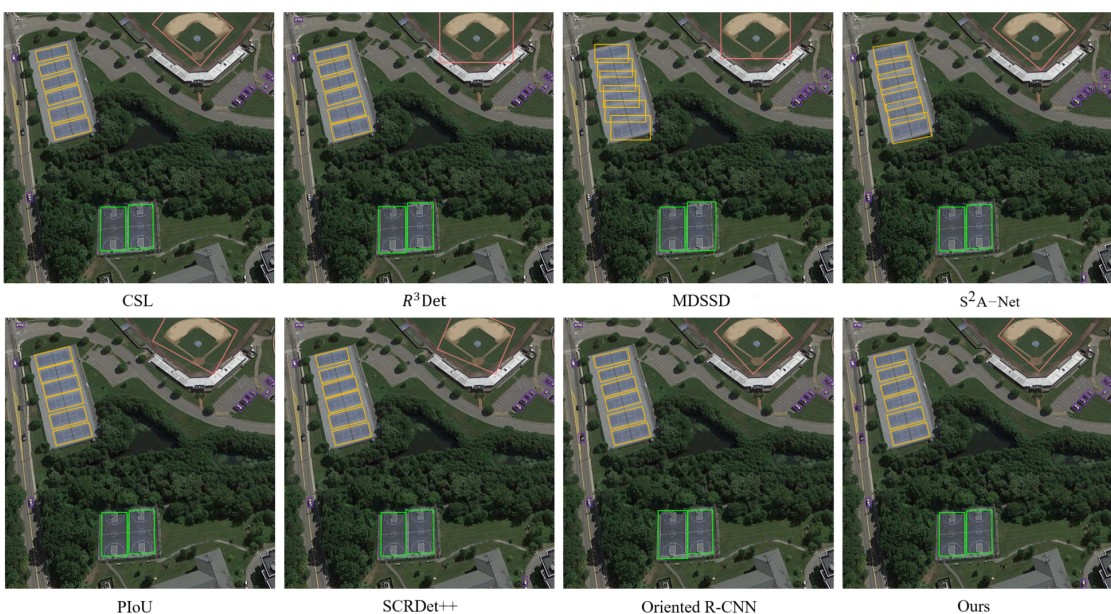

**Figure 12.** Visualization of the detection performance between our proposed method and the state-of-the-art methods for multi-scale object detection.

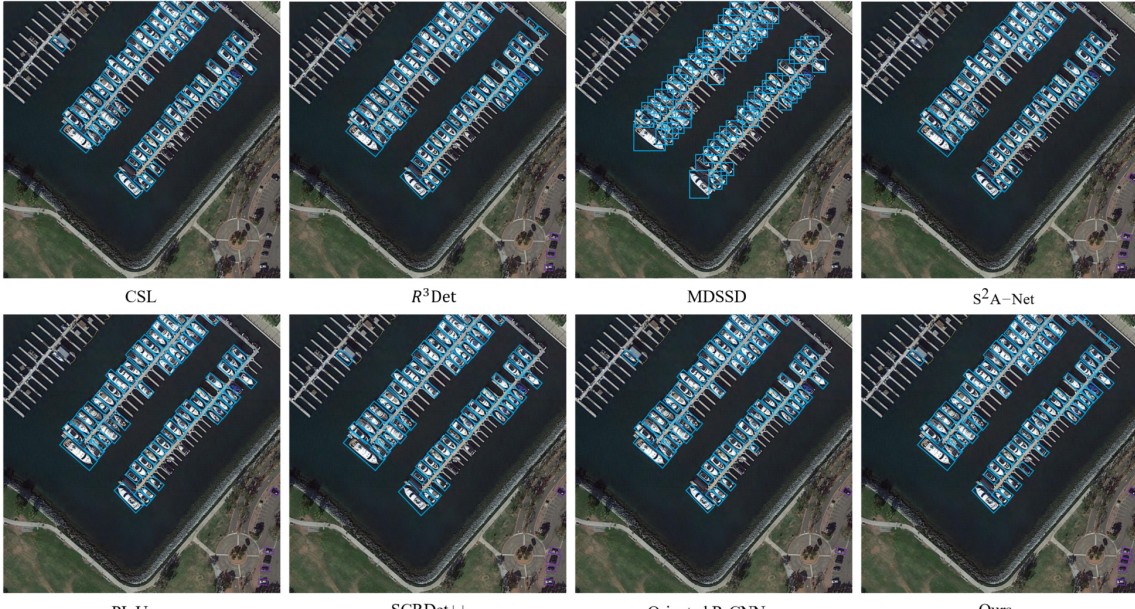

**Figure 13.** Visualization of the detection performance between our proposed method and the state-of-the-art methods for the detection of densely distributed small objects.

Figure 10 shows the detection results of a single type of object in the simple background. From the subjective point of view, the proposed method obtains the most accurate regression and classification, but the improvement effect is not great compared to other methods. This is because the scene is too simple, and general detection methods can achieve good detection results.

Figure 11 illustrates the detection results of different types of objects in complex backgrounds. From a subjective point of view, the proposed method achieves the lowest omission ratio and can detect small objects with little contrast with the background. In contrast, other methods suffer from varying degrees of missed detection. It can be inferred that the background has a great influence on the detection results, and the proposed

method can effectively solve this problem. This is attributed to the adoption of the PHDA module. The PHDA module suppresses the noise interference of complex backgrounds and improves the attention degree of foreground objects through the joint action of parallel channel attention and spatial attention, thus improving the detection accuracy of small objects in complex backgrounds effectively.

Figure 12 shows the detection results of multi-scale objects. From the subjective point of view, our detection results are the best. For a large-scale baseball diamond, our regression effect is the most consistent with the ground truth, and the prediction boxes of other methods are enlarged to varying degrees compared to the ground truth. For small-scale vehicles, the proposed method obtains the lowest omission ratio, and almost every vehicle can be detected. Generally, the proposed method achieves the best detection results for multi-scale objects in remote sensing images, and it does not miss the detection of small objects.

Figure 13 shows the detection results of densely distributed small objects. From a subjective point of view, the proposed method performs better than other methods in detecting dense small objects. The proposed method pays much attention to the densely distributed ships while considering the small cars in the non-dense area in the lower right corner. Our method achieves a lower omission ratio than other methods, and the regression effect is also better.

Figure 14 shows the detection results of the proposed method for multi-scale objects on high-resolution remote sensing images of large scenes. The proposed method achieves a good detection effect on small objects in high-resolution remote sensing images of large scenes, and it can accurately detect densely arranged cars, ships, or basketball courts and soccer ball fields with little distinction from the background. Interestingly, the enlarged image in the lower left corner shows that the proposed method also performs well for occluded car detection. Figure 15 shows the detection results of the proposed method on other remote sensing image examples.

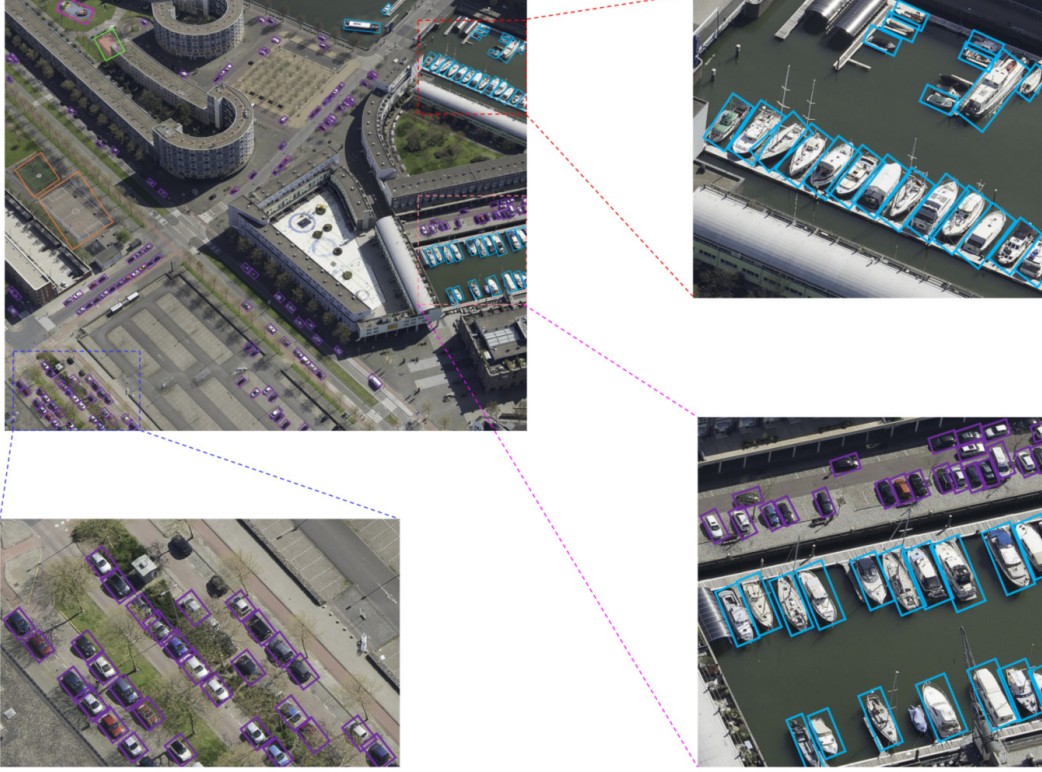

**Figure 14.** Visualization of the detection performance of our proposed method for multi-scale objects on high-resolution remote sensing images of large scenes.

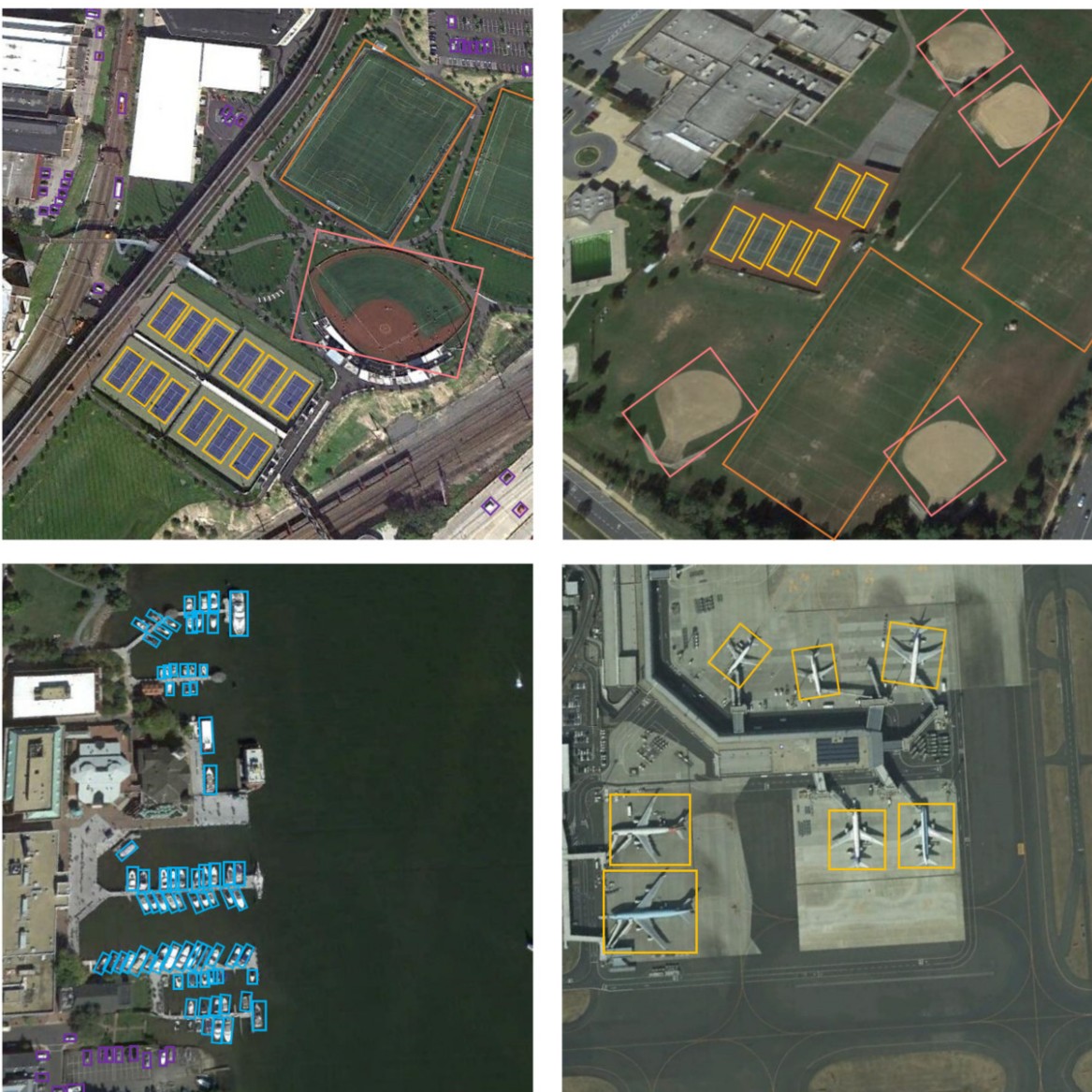

**Figure 15.** Visualization of the detection performance of our proposed method on other remote sensing image examples.

*4.5. Discussion*

The RSSO dataset is produced by filtering, cropping, and scaling images from three public datasets, namely NWPU VHR-10, DOTA, and UCAS AOD. Experiments are performed on the RSSO dataset, which makes the evolution of our proposed method more objective. Ablation experiments are performed on the proposed modules and the backbone network used in this paper, and the results verify the effectiveness of these modules and the backbone network. In comparison with other state-of-the-art methods, our method obtains some surprising results. It is found that our proposed method has a stronger detection ability for small objects and can effectively reduce the omission ratio of small objects in the case of simple backgrounds, complex backgrounds, multi-scale objects, and dense small objects. Meanwhile, although the detection speed of the proposed method is not the highest, it is still sufficient for practical engineering applications and can meet the requirements of real-time performance. The experimental results indicate that the proposed method can effectively improve the detection performance of small objects in remote sensing images. These satisfactory results are attributed to the reasonable architecture of our mode and

the numerous adjustments to the network model hyperparameters during the network training phase.

## 5. Conclusions

Detecting small objects in UAV optical remote sensing images suffers from low accuracy due to low contrast, dense distribution, and weak features. To solve this problem, this paper focuses on the influence of different levels of features and rotating bounding boxes on the detection performance of small objects and proposes a small object detection method based on the feature alignment of candidate regions for remote sensing images. Firstly, AFA-FPN defines the corresponding relationship between feature mappings, solves the misregistration of features between adjacent levels, and improves the recognition ability of small objects by aligning and fusing shallow spatial features and deep semantic features in convolutional neural networks. Secondly, the PHDA module captures local areas containing small object features through parallel channel domain attention and spatial domain attention, and it assigns a larger weight to these areas to eliminate the interference of background noise. Then, the rotation branch uses RROI to rotate the horizontal frame obtained by RPN, which avoids missing detection of small objects with dense distribution and arbitrary direction and prevents the object from not matching the characteristics of candidate regions. Next, the rotation branch uses RROI to rotate the horizontal box obtained by RPN, which solves the problem of missing detection of small objects with dense distribution and arbitrary direction and prevents feature mismatch between the object and candidate regions. Finally, the loss function is improved to better reflect the difference between the predicted value and the ground truth. Experiments are conducted on a self-made dataset RSSO, and the experimental results show that the proposed method can effectively enhance the representation ability of small objects, reduce the omission ratio of small objects, and achieve efficient and accurate detection of small objects in remote sensing images under different backgrounds.

In future work, the application of unsupervised models is the focus of our research. Unsupervised models can avoid the tedious labeling of datasets while improving the model's adaptivity. In addition, the use of lightweight models will be considered to improve model detection efficiency.

**Author Contributions:** Conceptualization, J.W., X.H. and F.S.; methodology and software, J.W. and F.S.; validation and formal analysis, J.W. and G.L.; resources and data curation, F.S.; writing original draft preparation, review, and editing, J.W., X.H. and F.S.; project administration, G.L.; funding acquisition, F.S. All authors have read and agreed to the published version of the manuscript.

**Funding:** This research was funded by the National Natural Science Foundation of China (grant number: 61671470), and the Key Research and Development Program of China (grant number: 2016YFC0802900).

**Institutional Review Board Statement:** Not applicable.

**Informed Consent Statement:** Not applicable.

**Data Availability Statement:** The data that support the findings of this study are available from the corresponding authors Faming Shao (shaofaming@aeu.edu.cn) and Guanlin Lu (luguanlin@aeu.edu.cn.com) upon reasonable request.

**Conflicts of Interest:** The authors declare no conflict of interest.

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
