# Peer review of "A Novel Method of Small Object Detection in UAV Remote Sensing Images Based on Feature Alignment of Candidate Regions"

_drones, doi:10.3390/drones6100292_

Round 1

Reviewer 1 Report (Previous Reviewer 3)

I have no more questions.

Author Response

Thank you for your review.

Reviewer 2 Report (New Reviewer)

In this paper, a novel method enhancing small object detection in UAV remote sensing images is introduced. Combined with Attention-based feature alignment FPN network, polarization hybrid domain attention and rotation branch technique, it is impressive that this cocktail method shows stronger detection ability for not only small but larger objects than the state-of-the-art techniques. The detection speed of about 24 FPS, which meet the real-time detection requirement, is appreciated as well. Yet, there are still critical shortcomings that make this work not to meet the criterion of the journal.

The most obvious drawback of this work is the lack of novel insight and methodology on small objects detection, which lower its academic importance. The core design of the detection algorithm seems to make only a slight modification on the basis of the existing algorithms (especially similar to Lu’s work in 2021[1]). In detail, The AFA-FPN method convoluted and fused features of the adjacent feature layers, which resembled the framework of the feature fusion SSD in [1]. Moreover, the core design of the PHDA is to calculate both channel attention and spatial attention branches, similar to the DAM algorithm in [1]. The most prominent difference between these two works is that Wang involved the rotation module for small objects in the whole algorithm, which became a regular procedure on small object detection.

In all, no matter how to improve this paper, the structure of the paper would be severely altered. Thus, I suggest the authors to reorganize this work and resubmit.

[1] Lu X, et al. ”Attention and feature fusion SSD for remote sensing object detection.” IEEE Transactions on instrument and measurement 70 (2021)

Author Response

First of all, thank you for your review. I have carefully studied the paper you mentioned[1]. From a macro point of view, your opinion seems reasonable, but the method of our paper is quite different from that of the paper you mentioned in micro details. According to your statement, as long as the paper uses the method of feature fusion and dual attention mechanism, then this paper is what you called “lack of novel insight and methodology” and “make only a light modification on the basis of the existing algorithms[1] “, then I can find out hundreds of similar papers. I am not the first person to adopt feature fusion and dual attention mechanism to improve the performance of the model, and I will never be the last, because it has been proved that they can indeed improve the performance of remote sensing small object detection.

Our proposed AFA-FPN can solve the misregistration between features of adjacent layers by predicting the flow field, ensure that the deep semantic features of small targets can be transferred to the shallow feature map, and effectively solve the problem of small target information loss during convolution. There is no similarity with the feature fusion module designed in this paper 1. And the dual attention mechanism designed by us is also different from the dual attention mechanism designed in Paper 1. We add polarization function behind the two attention branches to greatly suppress the noise interference of irrelevant areas to small target areas, which is much better than the dual attention mechanism designed in Paper 1. We also design a rotation branch according to the rotation characteristics of small objects to eliminate the mismatch between regional features and objects. The experimental results show that our proposed method can indeed improve the performance of remote sensing small object detection.

I can guarantee that our algorithm implementation has nothing in common with the paper you mentioned. I hope you can carefully read the contents of sections 3.1 and 3.2 of our paper and I'm sure you will change your mind after reading it.

Reviewer 3 Report (New Reviewer)

The paper presents a new solution to detect small objects from aerial images. The contribution consists of developing three modules to optimize the object detection process using attention: the Attention-based Feature Alignment FPN Module, Polarization Hybrid Domain Attention Module), and Rotation Branch. The paper presents a very interesting approach, but there are several issues to clarify and improve. 

First, there is a need for extensive proofreading because there are several language errors to fix, although the overall flow of the paper reads fine. 

From a technical perspective, here are the issues and concerns to consider. 

  • In the related works, the authors should add a summary table to make a qualitative comparison between related research, categorize them, and mention techniques used in other papers and how they differ from those used in this paper. 
  • In figure 1, it is unclear whether the modules execute in a cascade from left to right or RESNET runs in parallel with AFA-FPN. The relation between the two modules is not explained. RESNET performs classification, and it is not clear where the object detection module is. It seems RESNET is used for feature extraction, but the relation between colored layers should be explained. 
  • In Section 3.1, the authors presented the Attention-based Feature Alignment FPN Module. The idea seems interesting, but there are some ambiguous issues. Figure 2 shows no clear evidence of why the position attention module can be achieved with 1x1 convolution followed by GAP and Sigmoid. The text's explanation is unclear as the authors mentioned computing the position weight through GAP. How and which position is extracted for the image? Is the object detected at this stage of the module? Same thing for the upsampling clarification module. Basically, I have a problem with names assigned to the module and how they relate to the conv operations.
  • Also, what is the logic behind using elementwise multiplication inside the position attention?
  • Why not use elementwise multiplication instead of sum and vice-versa? What would be the impact on the AFA-FPN module? This will improve the understanding of the logic behind the module's design. 
  • In Figure 4, the authors used a branch with MP and AP and another with GMP and GAP. The authors should explain the relation between the pooling technique used and the type of attention associated with it, either channel or position. This relation is not clear. 
  • Figure 3 provides a good illustration of the process. The authors need to describe each feature map and how it relates to the designed attention module. 
  • There is a need to put the dimensions of the inputs and features maps at each layer in Figure 2, Figure 3, and Figure 4.
  • The experimental evaluation section presents a comparison between the baseline network and a combination of the different modules. This is good. However, the authors compared some SOTA techniques for small object detection. However, there is no comparison with traditional techniques such as Yolo (v4 and v7). A comparison with Yolov4/7 is needed in accuracy and FPS to see how the new methods improve over Yolo. 

Author Response

Response to Reviewer 3 Comments

Point 1: First, there is a need for extensive proofreading because there are several language errors to fix, although the overall flow of the paper reads fine.

Response 1: We have made language changes to the article, hoping to meet your requirements. The editorial certificate is shown in attached file.

Point 2: In the related works, the authors should add a summary table to make a qualitative comparison between related research, categorize them, and mention techniques used in other papers and how they differ from those used in this paper.

Response 2: We also think that a summary table can make readers understand the related algorithms more intuitively, and we have added a summary table at the end of the related work.

Point 3: In figure 1, it is unclear whether the modules execute in a cascade from left to right or RESNET runs in parallel with AFA-FPN. The relation between the two modules is not explained. RESNET performs classification, and it is not clear where the object detection module is. It seems RESNET is used for feature extraction, but the relation between colored layers should be explained.

Response 3: The arrow in figure 1 indicates that the modules are cascaded from left to right, instead of running in parallel. RESNET is to extract features, AFA-FPN and PHDA are used to process the features. Finally, the Ratation branch performs classification and regression of objects. Colored layers indicate that feature maps of the same color have the same size.

Point 4: In Section 3.1, the authors presented the Attention-based Feature Alignment FPN Module. The idea seems interesting, but there are some ambiguous issues. Figure 2 shows no clear evidence of why the position attention module can be achieved with 1x1 convolution followed by GAP and Sigmoid. The text's explanation is unclear as the authors mentioned computing the position weight through GAP. How and which position is extracted for the image? Is the object detected at this stage of the module? Same thing for the upsampling clarification module. Basically, I have a problem with names assigned to the module and how they relate to the conv operations.

Also, what is the logic behind using elementwise multiplication inside the position attention?

Why not use elementwise multiplication instead of sum and vice-versa? What would be the impact on the AFA-FPN module? This will improve the understanding of the logic behind the module's design.

Response 4: Each feature channel is related to different objects or backgrounds, with its feature activations at the corresponding position are activated. For example, if one channel is activated by the “person” class, the “person” located regions will be activated. More activated activations result in larger the GAP generated feature value, and thus the module pays more attention on this channel to hightlight these objects. That is, the resulting attention feature map focuses more on these positions of the attentioned objectt. The objects are highlighted, rather than detected at this module. That is the reason we call this module as the “position attention path” and how this module works in the model. We also follow your valuable suggestion to rename this module from “position attention module” to “object attention module” for better illustration.

For the upsampling calibration module. We perform a dynamic upsampling by learning the upsampling weights. The rational behind is that every pixel does not contribute equal in the upsampling procedure and we assign larger weights to those important pixels for better upsampling. For example, for the upsampling at the smooth regions, the conventional bilinear upsampling works well. But at those sharp regions, e.g., the object boundary regions, the  conventional bilinear upsampling usually fails to generate satisfactory results, while our upsampling calibration module will assign larger weights to the object or background pixels for better upsampling.

The convolutional operations are mainly used to control the feature map scales to adapt to the following attention or upsampling operations. If without the convolutional layer, the module training will be difficult because the input feature has to meet the requirements of different modules.

For the elementwise multiplication in the position attention module, we want to highlight those object-related feature channels, so we use the GAP and sigmoid functions to generate the channel weights and then multiply the weights to the feature map.

For the elementwise summation in the AFA-FPN module, we want to combine the input features, which are from different modules and are equally important.

In a word, the elementwise multiplication is for the attention module to highlight feature channels, and the elementwise summation is for the feature combination.

Point 5: In Figure 4, the authors used a branch with MP and AP and another with GMP and GAP. The authors should explain the relation between the pooling technique used and the type of attention associated with it, either channel or position. This relation is not clear.

Figure 3 provides a good illustration of the process. The authors need to describe each feature map and how it relates to the designed attention module.

Response 5: The “MP” and “AP” is for the position attention, and the “GAP”and “GMP” is for the channel attention.

For aggregating feature information, the AP/GAP is to learn the extent of the target object effectively or compute spatial statistics. The MP/GMP gathers another important clue about distinctive object features to infer finer channel-wise attention. Thus, we use both average-pooled and max-pooled features simultaneously.

Fi-1up is the feature map generated by the upsampling calibration path, where the object boundary is upsampled in a high quality.

Fi-1po is the feature map generated by the position attention path, where the objects are highlighted.

We have added this discussion in section 3.1.

Point 6: There is a need to put the dimensions of the inputs and features maps at each layer in Figure 2, Figure 3, and Figure 4.

Response 6: In Figure 2 and Figure 3, we don't fix which two layers of feature maps are used for the fusion operation.  and  only represent the adjacent feature maps, so the dimensions of each layer are not represented in the maps. Specifically, it corresponds to the top-down four characteristic diagrams of AFA-FPN module in Figure 2, and their dimensions are 2048, 1024, 512 and 256 respectively. We have shown the dimensions of the inputs and features maps at each layer in Figure 4.

Point 7: The experimental evaluation section presents a comparison between the baseline network and a combination of the different modules. This is good. However, the authors compared some SOTA techniques for small object detection. However, there is no comparison with traditional techniques such as Yolo (v4 and v7). A comparison with Yolov4/7 is needed in accuracy and FPS to see how the new methods improve over Yolo.

Response 7: We added this part in table 4 in section 4.4 , hoping to meet your requirements. Finally, thank you for your review and evaluation of our work. We have tried our best to revise your comments as required.

Round 2

Reviewer 2 Report (New Reviewer)

The   manuscript  has been revised according my comments,which can be accepted in current form.

Reviewer 3 Report (New Reviewer)

The authors addressed my comments. The paper can be accepted. 

This manuscript is a resubmission of an earlier submission. The following is a list of the peer review reports and author responses from that submission.

Round 1

Reviewer 1 Report

Very nice research with results and presentation. No further suggestions. Recommendation for publication as is. 

Author Response

感谢您对文章的审阅和对文章质量的肯定。我们感谢您的辛勤工作。

Reviewer 2 Report

1. Since there are too many mismatch and error in figures and formula, it is really hard to justify the proposed methods are correct or not.

For example, in Figure 2, flow field will performs element-wise add with bellow feature maps. However flow field deltai-1 has shape Hi-1*Wi-1*2 while other feature maps have shape Hi*Wi*C.

In Equation (3), left part Pi and right part has different shape. So that they will not be equal.

In Equation (4), Fi is a feature map in previous definition but it is used to be a function in this equation. Also, how can Fi be applied to Pi and Pi-1 at the same time? 

From equation (2) to (4), do the authors want to perform deformable convolution? If yes, the author could cite it and describe it in the paper. If no, the author should provide correct description and formula.

From Equation (6), Mi is applied to Fi', however in Figure 2, Mi is applied to theta(Fi').

In Equation (8), how the author concatenate c3*1 and c1*3? I think the author want to concatenate c3*1(Tmp) and c1*3(Tmp). Also, Equation (8) and Figure 4 are mismatch.

Equation (9) and Figure 4 are mismatch, Equation (9) uses element-wise multiplication to perform attention mechanism, while Figure 4 uses tensor product to perform attention mechanism.

RRoI, FL, and CIoU are proposed by other researchers, so I do not check these equations. But in Equation (21), there are only losses for classification and regression of final output. Does RPN has no loss?

2. All of experiments are performed on self-made RSSO dataset. The RSSO dataset is mainly composed by small objects in UAV scene. However, In real application of UAV remote sensing scenario, there will contains various size of objects. Although RSSO dataset can show the proposed methods work very well on small objects, people will want to know if the system can work well in real scenario. It is better to run experiments on DOTA dataset too, and show how much performance enhanced on small objects and how much performance drop on large objects.

3. CurrentlyThe author should describe what is the benefit of their proposed methods. For example, why Tap branch uses 3*3 convolution but Tmp uses both 3*1 and 1*3 convolution.

4. Baseline-FPN-CIoU gets similar performance as Baseline-AFAFPN-GIoU and final modell applied CIoU. It is better to show results of Baseline-AFAFPN-CIoU. This can help people know actual performance gain from proposed AFAFPN.

5. Bad formatting. There are many tables are split into different pages. Table 2 is hard to read.

Author Response

Point 1: 1. Since there are too many mismatch and error in figures and formula, it is really hard to justify the proposed methods are correct or not.

For example, in Figure 2, flow field will performs element-wise add with bellow feature maps. However flow field deltai-1 has shape Hi-1*Wi-1*2 while other feature maps have shape Hi*Wi*C.

In Equation (3), left part Pi and right part has different shape. So that they will not be equal.

In Equation (4), Fi is a feature map in previous definition but it is used to be a function in this equation. Also, how can Fi be applied to Pi and Pi-1 at the same time?

From equation (2) to (4), do the authors want to perform deformable convolution? If yes, the author could cite it and describe it in the paper. If no, the author should provide correct description and formula.

From Equation (6), Mi is applied to Fi', however in Figure 2, Mi is applied to theta(Fi').

In Equation (8), how the author concatenate c3*1 and c1*3? I think the author want to concatenate c3*1(Tmp) and c1*3(Tmp). Also, Equation (8) and Figure 4 are mismatch.

Equation (9) and Figure 4 are mismatch, Equation (9) uses element-wise multiplication to perform attention mechanism, while Figure 4 uses tensor product to perform attention mechanism.

RRoI, FL, and CIoU are proposed by other researchers, so I do not check these equations. But in Equation (21), there are only losses for classification and regression of final output. Does RPN has no loss?

Response 1: There are many questions about mathematical equations, and I try my best to explain each question clearly. In fact, like most depth study papers, our mathematical equations are more inclined to express the logic of our methods. Therefore, the mathematical equations are not complete enough to reproduce our methods through mathematical equations. We emphasize the complementarity of figures and formula s, not the consistency.

For example, you mentioned the problem of inconsistent shapes of Figure 2. In the process of feature fusion, the fused features of different depths must have different scales, but in the paper, there is no need to explain in detail how to fuse these two features of different scales. In fact, in common experiment environments, whether matlab or python, they all use a simple function to achieve feature fusion of different scales. Without the optimization of fusion strategy, we think it is unnecessary to make an explanation.

What we are trying to express in equation (3) is not that the left and right sides are equal, but that the right side is assigned to the left side. The problem of unequal shapes is solved by scaling function, and we don't think it is necessary to express it in detail in the formula. We think that the readability of this expression is in line with readers' requirements.

The definition of Fi in equation (4) is still a feature map, and P is a point on the feature map. This expression does not mean a function, but the corresponding relationship between the pixels of the deep feature map and the shallow feature map. In addition, we recognize that there are some problems in equation (4), and we have updated equation (4). Thank you for your careful review of the article, hoping to meet your requirements.

It's not a deformable convolution. We have already answered the question of inconsistent shapes above and improved the formula.

Mi is the attention mapping of the output, and theta (Fi') is included in it. Therefore, Mi is not applied to theta (Fi'). At the same time, we can also see from Figure 2 that the attention branch adopts the residual structure, and Mi is applied to Fi' to output the final result of the attention branch.

It is true that we did not show the microstructure of spatial attention in Figure 4, but this does not mean that it is impossible to concatenate c3*1 and c1*3. Moreover, we don't think that showing all the microscopic details of the structure in Figure 4 will greatly improve the quality of the article, but will lead to redundancy. In order to reduce the parameters and speed up the training of the average pooling with low weight in spatial attention, we use asymmetric 3*1 and 1*3 convolution to compress and speed up the model. The figures only plays an auxiliary role to the formulas, but can't fully express the formulas. We emphasize the complementarity of figures and formulas rather than consistency.

We recognized the mismatch between equation 9 and Figure 4, and modified Figure 4.

After your reminder, we are also aware of this problem. In the experiment, we used the classic RPN loss proposed by Faster RCNN, but it was not reflected in the manuscript. It was a mistake in our work, and we have revised it.

Point 2: All of experiments are performed on self-made RSSO dataset. The RSSO dataset is mainly composed by small objects in UAV scene. However, In real application of UAV remote sensing scenario, there will contains various size of objects. Although RSSO dataset can show the proposed methods work very well on small objects, people will want to know if the system can work well in real scenario. It is better to run experiments on DOTA dataset too, and show how much performance enhanced on small objects and how much performance drop on large objects.

Response 2: The DOTA data set contains many remote sensing images of large scenes, which are quite different from the images taken by drones. In order to fit the research direction of the journal, we selected the drone images from three public data sets including DOTA data set or the images taken by carriers at the same shooting height as drones for experimental training and testing. Moreover, our database also contains targets of different scales, and classifies the targets of different scales. The experimental results are shown in Table 3. It can be seen that the detection effect of small scale targets is obviously improved by our proposed method, and other scale targets are slightly improved. Subjectively, the results obtained from experiments can be analyzed and compared, and our proposed method has obvious advantages in detecting small targets in UAV remote sensing images.

Point 3: CurrentlyThe author should describe what is the benefit of their proposed methods. For example, why Tap branch uses 3*3 convolution but Tmp uses both 3*1 and 1*3 convolution.

Response 3: We believe that your suggestions can play a key role in improving the quality of articles, and have made the following improvements and explanations.

The Maximum pooling is to maximize the pixels in the pool area, and the feature map obtained in this way is more sensitive to texture feature information. Average pooling is to average the images in the pool area, and the feature information obtained in this way is more sensitive to the background information. Spatial attention is mainly focused on extracting the spatial position weight of the input feature pixels, which does not require high background accuracy. Therefore, asymmetric 3*1 and 1*3 convolution are used to compress and accelerate the model in average pooling to reduce the number of parameters. For maximum pooling, we use traditional 3*3 convolution to extract texture feature information.

  We also checked the method innovations in other places, and explained the benefit of other method innovations, hoping to meet your requirements.

Point 4: Baseline-FPN-CIoU gets similar performance as Baseline-AFAFPN-GIoU and final modell applied CIoU. It is better to show results of Baseline-AFAFPN-CIoU. This can help people know actual performance gain from proposed AFAFPN.

Response 4: Your suggestion is very pertinent, and it can make readers know more about the comparison of ablation experiment results. However, we consider that the length of ablation experiment in academic papers is not too long, and we can get the true performance gain of AFA-FPN to the algorithm framework by comparing the results in the first and second rows in Table 1.

Point 5: Bad formatting. There are many tables are split into different pages. Table 2 is hard to read.

Response 5: After your reminder, we also think that the format of Table 2 is not easy for readers to read. We have modified it and checked the formats of other tables and figures to ensure that it is easy for readers to read.

Reviewer 3 Report

This paper proposes a small object detection framework for remote sensing images. Overall, the methods and experiments are completed and the manuscript quality is good. However I still have the following concerns:

1.       In the Introduction part, since this paper is submitted to Drones, is there any difference in the small object detection between UAV and satellite remote sensing?

2.       Line 167: the intention of building the RSSO dataset should be briefly introduced here, although readers can see relative description in the experiment section.

3.       Sect. 3.1: is the AFA-FPN an originally designed module? It is suggested to qualitatively introduce the novel idea of AFA-FPN first, e.g., to solve what problem by predict the flow field, and then to describe the details.

4.       Sect. 3.2: is the PHDA a novel module or is it an existing method and just adopted in this framework? Since the purpose of attention modules are similar, what is the advantage or characteristic compared with other attention modules? The reason why you design it like this or why PHDA is adopted should be explained.

5.       Line 383: GioU, CioU, capital letters.

6.       Sect. 4.1, about the dataset: The datasets included in this paper are not typical UAV data. For example, VHR-10 contains images from Google Earth and Vaihingen data (satellites). DOTA contains images from various sensors and platform such as Google Earth, JL-1 satellite, ZY and GF satellites. Since this journal is about Drones, please seriously consider the consistency of title, purpose and journal scope of this research.

7.       Table 1: HDA or PHDA?

8.       Figure 8: it is suggested that Fig. 8 and its description can be put behind Table 2, or the method names of 1-7 should be listed in the legend. Or readers must skip to Table 2 and look it back.

9.       Table 2: if there are too many subclasses, the method names should be listed again in the lower part. By the way, “mothed” is a wrong spelling in the table.

Author Response

Thanks for reviewing this article. We greatly appreciate your work and have responded to all your comments.

Point 1: In the Introduction part, since this paper is submitted to Drones, is there any difference in the small object detection between UAV and satellite remote sensing?

Response 1: Simply put, satellite remote sensing can see far because it flies high, so its observation information is macroscopic and comprehensive, which is its most important feature; In addition, it can be continuously observed for a long time to form time series information; Generally speaking, the cost of remote sensing satellites is lower than that of drones, and it is not limited by natural conditions (such as no man's land, no man's island).

Because the UAV is near the ground, its image resolution is generally high, and it is not affected by the weather (usually, cloud-free weather is required for satellite remote sensing). It has good timeliness and can shoot video images, but its observation range is small, its cost is relatively high, and its data processing is complicated.

Point 2: Line 167: the intention of building the RSSO dataset should be briefly introduced here, although readers can see relative description in the experiment section.

Response 2: We think your suggestion is very pertinent, and we have introduced the intention of establishing RSSO dataset in this part.

Point 3: Sect. 3.1: is the AFA-FPN an originally designed module? It is suggested to qualitatively introduce the novel idea of AFA-FPN first, e.g., to solve what problem by predict the flow field, and then to describe the details.

Response 3: The AFA-FPN is an originally designed module. In response to your suggestion, we made an improvement and made a qualitative analysis of AFA-FPN before introducing the details of the algorithm in Sect. 3.1.

Point 4: Sect. 3.2: is the PHDA a novel module or is it an existing method and just adopted in this framework? Since the purpose of attention modules are similar, what is the advantage or characteristic compared with other attention modules? The reason why you design it like this or why PHDA is adopted should be explained.

Response 4: The PHDA is a novel module. The general attention mechanism does not involve polarization function, only gives a cursory attention to the target that needs attention. For the detection of small objects in UAV images, the general attention mechanism can't achieve good results because of their small size, various types and different directions. Therefore, we introduce the polarization function. It can be seen from the expression formula 10 that the irrelevant areas with attention weight less than 0.5 are greatly suppressed to reduce the noise interference to small object areas, while the small object areas with attention weight greater than 0.5 are hardly suppressed. The introduction of the polarization function further enhances the effect of attention mechanism and achieves the purpose of highlighting the characteristic areas of small objects.

Thank you for your suggestion, which is of great help to improve the quality of the article. We have explained in Scet. 3.2 why we designed the PHDA.

Point 5: Line 383: GioU, CioU, capital letters.

Response 5: We have made corrections and checked the capitalization of other places.

Point 6: Sect. 4.1, about the dataset: The datasets included in this paper are not typical UAV data. For example, VHR-10 contains images from Google Earth and Vaihingen data (satellites). DOTA contains images from various sensors and platform such as Google Earth, JL-1 satellite, ZY and GF satellites. Since this journal is about Drones, please seriously consider the consistency of title, purpose and journal scope of this research.

Response 6: We know that the selected datasets do not only contain UAV images, so we have screened and combined these datasets. As you can see from the examples, the images we selected are all aerial images of UAVs or images taken by other carriers at the same height as UAVs, which is of real value for the research of UAV remote sensing small object detection.

Point 7: Table 1: HDA or PHDA?

 Response 7: PHDA. Our carelessness led to this mistake. We have corrected this. Thank you for reminding me. You can find such minor mistakes, which shows your careful attitude. Thank you again for reviewing this manuscript.

Point 8: Figure 8: it is suggested that Fig. 8 and its description can be put behind Table 2, or the method names of 1-7 should be listed in the legend. Or readers must skip to Table 2 and look it back.

 Response 8: In order to highlight the good effect of each of our modules on improving the detection performance of small objects, this paper compares the P-R curves of ablation experiments for detecting small, medium and large objects and all objects. The comparison results are shown in fig. 8, in which serial numbers 1 to 7 respectively represent the first to seventh rows corresponding to the ablation experiment table 1.

Point 9: Table 2: if there are too many subclasses, the method names should be listed again in the lower part. By the way, “mothed” is a wrong spelling in the table.

 Response 9: After your reminder, Table 2 is really inconvenient for readers to read, and we have modified it.

Round 2

Reviewer 2 Report

1. The author have not checked and corrected figures and equations carefully. These are not only the consistency of figures and equations, but also the logic of the proposed methods.

1-1. From line 218, we can know F_{i} and F_{i-1} are deep and shallow feature maps respectively. That means resolution of F_{i} is smaller than F_{i-1}.

1-2. Since there are no notations on Figure 2, according to 1-1, we assume green feature map is F_{i} and blue feature map is F_{i-1}.

1-3. F_{i}^{'} in Equation (1) has same shape as F_{i-1}.

1-4. Equation (2), (3), and (4) are unreadable. For example, P_{i} in these three equations seems have different meaning. In Equation (3), P_{i} seems has same resolution as F_{i}^{'} and F_{i-1}. However, in Equation (4), P_{i} seems has same resolution as F_{i}. Moreover, I do not think the flow field performs element-wise add with other feature maps directly as shown in Figure 2, I think the flow field is used to sample some points on some feature maps to generate a feature map for feature alignment.  Also, I do not think the flow field is applied to pixels in F_{i} and F_{i-1} as shown in Equation (4).

1-5. In Equation (5), F_{i}^{'} is passing through GAP and sigmoid functions. However, in Figure 2, F_{i}^{'} will first pass through an 1x1 convolution then pass through GAP and sigmoid functions.

1-6. Equation (6) has same problem as Equation (5) mentioned in 1-5.

1-7. In Figure 3, resolution of F_{i} is larger than F_{i-1}, which is conflict of line 218 and Equation (1). More details are shown in 1-1 and 1-3. This also cause the Figure 3 becomes unreasonable.

2. The are some typos in the draft. For example, somewhere PHDA are used, somewhere HDA are used.

3. From the results, we can find that the proposed method with GIoU loss has lower AP than other baseline methods. That is one of the reason why I suggest to provide more ablation studies based on CIoU loss.  It is better to compare proposed method with other similar methods. For example IoU-based losses designed for oriented objects, such as PIoU[a], feature alignment methods for oriented objects, such as S^{2}A-Net[b], and state-of-the-art methods in this field, such as Oriented R-CNN[c].

[a] Chen, Z., Chen, K., Lin, W., See, J., Yu, H., Ke, Y., & Yang, C. (2020, August). Piou loss: Towards accurate oriented object detection in complex environments. In European conference on computer vision (pp. 195-211). Springer, Cham.

[b] Han, J., Ding, J., Li, J., & Xia, G. S. (2021). Align deep features for oriented object detection. IEEE Transactions on Geoscience and Remote Sensing60, 1-11.

[c] Xie, X., Cheng, G., Wang, J., Yao, X., & Han, J. (2021). Oriented R-CNN for object detection. In Proceedings of the IEEE/CVF International Conference on Computer Vision (pp. 3520-3529).

Author Response

Point 1: 1. The author have not checked and corrected figures and equations carefully. These are not only the consistency of figures and equations, but also the logic of the proposed methods.

1-1. From line 218, we can know F_{i} and F_{i-1} are deep and shallow feature maps respectively. That means resolution of F_{i} is smaller than F_{i-1}.

1-2. Since there are no notations on Figure 2, according to 1-1, we assume green feature map is F_{i} and blue feature map is F_{i-1}.

1-3. F_{i}^{'} in Equation (1) has same shape as F_{i-1}.

1-4. Equation (2), (3), and (4) are unreadable. For example, P_{i} in these three equations seems have different meaning. In Equation (3), P_{i} seems has same resolution as F_{i}^{'} and F_{i-1}. However, in Equation (4), P_{i} seems has same resolution as F_{i}. Moreover, I do not think the flow field performs element-wise add with other feature maps directly as shown in Figure 2, I think the flow field is used to sample some points on some feature maps to generate a feature map for feature alignment.  Also, I do not think the flow field is applied to pixels in F_{i} and F_{i-1} as shown in Equation (4).

1-5. In Equation (5), F_{i}^{'} is passing through GAP and sigmoid functions. However, in Figure 2, F_{i}^{'} will first pass through an 1x1 convolution then pass through GAP and sigmoid functions.

1-6. Equation (6) has same problem as Equation (5) mentioned in 1-5.

1-7. In Figure 3, resolution of F_{i} is larger than F_{i-1}, which is conflict of line 218 and Equation (1). More details are shown in 1-1 and 1-3. This also cause the Figure 3 becomes unreasonable.

Response 1: We sincerely thank reviewers for their valuable comments! We carefully revised the AFA-FPN section and optimized the figures for better presentation, as suggested by the reviewers.

Point 2: The are some typos in the draft. For example, somewhere PHDA are used, somewhere HDA are used.

Response 2: We have revised it and checked the full text to avoid typos. Thank you for your careful review of the article.

Point 3: From the results, we can find that the proposed method with GIoU loss has lower AP than other baseline methods. That is one of the reason why I suggest to provide more ablation studies based on CIoU loss.  It is better to compare proposed method with other similar methods. For example IoU-based losses designed for oriented objects, such as PIoU[a], feature alignment methods for oriented objects, such as S^{2}A-Net[b], and state-of-the-art methods in this field, such as Oriented R-CNN[c].

[a] Chen, Z., Chen, K., Lin, W., See, J., Yu, H., Ke, Y., & Yang, C. (2020, August). Piou loss: Towards accurate oriented object detection in complex environments. In European conference on computer vision (pp. 195-211). Springer, Cham.

[b] Han, J., Ding, J., Li, J., & Xia, G. S. (2021). Align deep features for oriented object detection. IEEE Transactions on Geoscience and Remote Sensing, 60, 1-11.

[c] Xie, X., Cheng, G., Wang, J., Yao, X., & Han, J. (2021). Oriented R-CNN for object detection. In Proceedings of the IEEE/CVF International Conference on Computer Vision (pp. 3520-3529).

Response 3: We show the results of Baseline-AFAFPN-CIoU in the ablation studies, which can help people know actual performance gain from proposed AFAFPN. At the same time, we also think that adding comparative experiments with the three methods you mentioned can add a lot to the article, and we have cited them in related work. However, due to the limitation of the revision deadline (8.21) and the length of the article, we consider not to increase the comparative experiment in this article. We will definitely consider your suggestions in future research to conduct experimental comparisons with these state-of-the-art methods.

If you must request to add a comparative experiment, can the editor give us more time for the experiment, so that we can complete the experiment and get the desired results.

Reviewer 3 Report

Most of my questions has been answered.

There still has some writing problem such as in Table 3: "Mothed".

Please check the writings of the whole manuscript before publication to avoid other typos.

Author Response

Point 1: There still has some writing problem such as in Table 3: "Mothed".

Please check the writings of the whole manuscript before publication to avoid other typos.

Response 1: We have revised it, checked the full text, and revised the typos that appeared. Thank you for your careful review of the article.
